



# The tidal effects in the Finite-volumE Sea ice–Ocean Model (FESOM2.1): a comparison between parameterised tidal mixing and explicit tidal forcing

Pengyang Song[1], Dmitry Sidorenko[1], Patrick Scholz[1], Maik Thomas[2,3], and Gerrit Lohmann[1,4]

[1]Alfred Wegener Institute, Helmholtz Centre for Polar and Marine Research (AWI), Bremerhaven, Germany
[2]Helmholtz Centre Potsdam–GFZ German Research Centre for Geosciences, Potsdam, Germany
[3]Institute of Meteorology, Freie Universität Berlin, Berlin, Germany
[4]MARUM–Centre for Marine Environmental Sciences, University of Bremen, Bremen, Germany

**Correspondence:** Pengyang Song (pengyang.song@awi.de)

**Abstract.** Tides are proved to have a significant effect on the ocean and climate. Previous modelling research either adds a tidal mixing parameterisation or an explicit tidal forcing to the ocean models. However, no research compares the two approaches in the same framework. Here we implement both schemes into a general ocean circulation model and assess both methods by comparing the results. The aspects for comparison involve hydrography, sea ice, mixed layer depth, Meridional Overturning Circulation (MOC), vertical diffusivity, barotropic streamfunction and energy diagnostics. We conclude that although the mesh resolution is poor in resolving internal tides in most mid-latitude and shelf-break areas, explicit tidal forcing still shows stronger tidal mixing at the Kuril–Aleutian Ridge and the Indonesian Archipelago than the tidal mixing parameterisation. Beyond that, the explicit tidal forcing method leads to a stronger upper cell of the Atlantic MOC by enhancing the Pacific MOC and the Indonesian Throughflow. Meanwhile, the tidal mixing parameterisation leads to a stronger lower cell of the Atlantic MOC due to the tidal mixing in deep oceans. Both methods maintain the Antarctic Circumpolar Current at a higher level than the control run by increasing the meridional density gradient but with different mechanisms. We also show several phenomena that are not considered in the tidal mixing parameterisation, for example, the changing of energy budgets in the ocean system, the bottom drag induced mixing on the continental shelves, and the sea ice transport by tidal motions. Due to the limit of computational capacity, an internal-tide-resolving simulation is not feasible for climate studies. However, a high-resolution short-term tidal simulation is still required to improve parameters and parameterisation schemes in climate studies.

## 1 Introduction

Based on Sandström's theorem, the ocean is proved not to be a heat engine (e.g., Huang, 2009). That is to mean, the global Thermohaline Circulation (THC) is not driven by the heat difference between the equator and the poles. Lot has been discussed on what is driving the global THC. Munk and Wunsch (1998) concluded that tides and winds maintain the global THC and ocean stratification. They also estimated the global mean vertical diffusivity (the same as diapycnal diffusivity in the whole context) as $10^{-4}$ m$^2$ s$^{-1}$, which mismatches $10^{-5}$ m$^2$ s$^{-1}$ from observational results in most parts of the global ocean. Thus, some areas with large vertical diffusivities are expected to contribute to the global strength of upwelling. Observational results





(Polzin et al., 1997; Ledwell et al., 2000) present large vertical diffusivities at near-bottom rough topographies in the Mid-Atlantic Ridge, which is believed to be generated by tide–topography interactions. These observational results indicate the

significance of tide-induced mixing effects from a climatic point of view.

Tide-induced mixing has long been considered in modelling research. Plenty of research on tuning vertical diffusivity parameters has been conducted to reduce model biases. By testing different vertical diffusivity parameters in an ocean general circulation model (OGCM), Bryan (1987) found that ocean circulation is sensitive to vertical diffusivity. As vertical diffusivity increases, the thickness of ocean thermocline, the strength of meridional overturning circulation (MOC), and poleward heat

transport increase. After the bottom-enhanced structure of vertical diffusivities is revealed from the observations (Polzin et al., 1997), the vertical diffusivity in ocean models is now considered as a vertical profile and not anymore as a constant parameter. Tsujino et al. (2000) discussed the influence of different vertical diffusivity profiles on the global ocean circulations, concluding that different profile shapes lead to different strengths of Pacific MOC (PMOC). However, most early research does not consider the horizontal inhomogeneity of tide-induced mixing in the global ocean. Since the raising of a tide-induced mixing

parameterisation (St. Laurent et al., 2002; Simmons et al., 2004), a better treatment that considers both vertical variation and horizontal inhomogeneity of the vertical diffusivity is widely used in OGCMs. It should be noted here that the horizontal distribution of the enhanced vertical diffusivities is related to the baroclinic tide conversion rate, which is provided by an analytical or numerical tidal model (e.g., Jayne, 2009). Thus the horizontal distribution is closer to the real ocean. Following the Community Ocean Vertical Mixing (Griffies et al., 2015, CVMIX) library, hereinafter, we name this parameterisation as CVMIX_TIDAL.

In the following research, CVMIX_TIDAL, along with parameterisations with similar formulas (e.g., Friedrich et al., 2011; Tatebe et al., 2018), has been adopted in OGCMs to reveal the tidal effect on the ocean and climate. It has been proved that the consideration of tide-induced mixing may have an impact on aspects related to the global THC, for example, the Atlantic MOC (AMOC) (Simmons et al., 2004; Jayne, 2009), the strength and deepness of Antarctic Circumpolar Current (ACC) (Saenko and Merryfield, 2005), the Southern Ocean sea ice, and the Circumpolar Deep Water formation (Tatebe et al., 2018), the ventilation

of the North Pacific Intermediate Water and the deep North Pacific circulation (Oka and Niwa, 2013), and the ocean biogeochemistry (Friedrich et al., 2011). Lee et al. (2006) also implemented bottom-enhanced vertical diffusivities in an OGCM. However, the main difference from above is that they enhanced vertical diffusivities on continental shelves, parameterising the tidal shear caused by the bottom drag in shallow areas. Their results show that the mixing caused by tidal shear on continental shelves improves the modelled hydrography at coastal areas and adjacent marginal seas.

Instead of adding parameterisation schemes to OGCMs, another way to consider the tidal effect is to add an explicit tidal forcing to the momentum equations. Global tides have been added to some OGCMs (e.g., Thomas et al., 2001; Shriver et al., 2012; Arbic et al., 2018; Logemann et al., 2021), but little research has been conducted from a climatic point of view. By comparing sensitivity runs with and without tidal motions in an OGCM, Schiller (2004) found that tidal motions can rectify Indonesian Archipelago hydrography and the Indonesian Throughflow (ITF) transport. Similar work via a coupled climate

model concludes that the simulation with tidal motions can improve the modelled North Atlantic Current and Western European Climate (Müller et al., 2010). Yu et al. (2016) also implemented a tidal module into an OGCM. But unlike the above research that concentrates on regional circulations, they discuss the influence on global hydrography and circulation. The main finding





is that the tidal motions reduce the strength of wind-induced circulation and the upper cell of AMOC while slightly increasing
the strength of the bottom cell of AMOC. In addition, a special kind of explicit forcing representing the tidal effect is the tidal
residual circulation. By considering the tidal residual circulation as an external forcing in an OGCM, Bessières et al. (2008)
concluded that the ocean state is insensitive to the tidal residual circulation, which can be neglected in climate research.

Generally speaking, there are mainly two approaches to study the tidal effect on the ocean and climate: parameterised tidal
mixing and explicit tidal forcing. Even though both approaches have been conducted in previous research, no research has
assessed these two approaches in the same framework. Therefore, we implement both approaches in our work. Our objectives
are:

(1) to find out the differences between these two approaches in expressing the effect of internal tide mixing in an OGCM;

(2) to evaluate the advantages and disadvantages of these two approaches in climate research;

(3) to see if there are any tidal effects other than internal tide mixing that would change the model results, which is not
considered in the CVMIX_TIDAL parameterisation.

In Sect. 2, we first introduce the applied numerical model and its configuration. Then in Sect. 3, we verify our newly
implemented tidal potential model. The model results are presented in Sect. 4 and are further discussed in Sect. 5. The last
section is the conclusion of this work.

## 2  Model description

The model applied in this work is the Finite-volumE Sea ice–Ocean Model (Danilov et al., 2017, FESOM2), which is the
successor of the Finite Element Sea ice–Ocean Model (Wang et al., 2014, FESOM). FESOM2 applies unstructured meshes,
which are composed of irregular-sized triangles. In FESOM2 meshes, scalars and vectors are located at triangle vertices and
triangle centres, respectively. By using multi-resolution meshes, specific regions of the global ocean can be well studied with
moderate computational effort. The arbitrary Lagrangian–Eulerian vertical coordinate, which incorporates different vertical
coordinates in the same frame, is introduced in FESOM2 (Danilov et al., 2017; Scholz et al., 2019). In this work, we apply the
latest release of the model, namely, FESOM2.1. Thus in this paper, all terminologies for FESOM2 refer to FESOM2.1.

### 2.1  Model setup

In this work, we apply a common mesh for FESOM2, namely, the CORE-II mesh (https://fesom.de/models/meshessetups/).
The CORE-II mesh features low resolution ($\sim 1°$) in mid-latitude areas and high resolution at coastal regions, the equator, and
north of $50°$ N. It has about 127k horizontal vertices and has 47 progressively thickening vertical layers from top to bottom.
Since our work includes the simulation of tidal motions, the instant sea surface displacement can exceed 10 m at some spots,
which requires either to thicken the uppermost vertical layer (from 5 m to 20 m) or to select an appropriate vertical coordinate.
Here we keep the vertical layer configuration while changing the vertical coordinate to the z-star coordinate. To keep a positive
thickness for the uppermost layer, the z-star coordinate distributes ocean surface displacement into changes in layer thicknesses
for all vertical layers except the bottom layer. A standard timestep for FESOM2 configuration is 2700 s, which is too long to





resolve fast-travelling barotropic tides. Through sensitivity tests, an optimal timestep of 720 s has been determined (see Sect.
3). In addition, since the model considers tidal motions and some related internal gravity waves, the Courant–Friedrichs–Lewy
(CFL) stability condition could be constrained by the vertical advection term (Lemarié et al., 2015). Thus, a vertical velocity
splitting method is applied in this work to keep vertical advection and oceanic mixing stable (Shchepetkin, 2015; Danilov et al.,
2017). Instead of explicitly treating the whole vertical advection term, when the vertical Courant number exceeds unity, the

excess part of the vertical velocity would be treated implicitly.

Except for the customised model configuration above, we adopt other generic settings for FESOM2 in this work. The model
starts from the Polar Science Centre Hydrographic Climatology (Steele et al., 2001, PHC3) and is driven by the Coordinated
Ocean-ice Reference Experiments phase II (CORE-II) reanalysis atmospheric forcing (Large and Yeager, 2009) ranging from
1948 to 2009. In addition, the sea surface salinity is restored to climatology values during the simulation. The horizontal

viscosity in the model applies an ocean kinetic energy backscatter parameterisation scheme (Juricke et al., 2019). Constant
background mixing is set as $10^{-4}$ and $10^{-5}$ m$^2$ s$^{-1}$ for vertical viscosity and diffusivity. In addition to the constant background
mixing, we apply the K-profile parameterisation (Large et al., 1994, KPP) in this work. The Gent–McWilliams scheme (Gent
and Mcwilliams, 1990) and the Redi isoneutral diffusion (Redi, 1982) are applied to the parameterisation of subgrid eddy-
induced mixing. Note here that the Gent–McWilliams parameterisation applied is formulated following Ferrari et al. (2010).

## 2.2 The tidal potential module

We implement a global tidal potential module (Thomas et al., 2001) into FESOM2 to consider the real tidal motions. The tidal
potential in FESOM2 can be expressed as

$$\Omega = \Omega_g + \Omega_e + \Omega_{SAL} = \alpha\Omega_g + \beta g\eta. \tag{1}$$

The tidal forcing is expressed in the model as the gradient of the tidal potential $\Omega$, which consists of three parts according to Eq.

(1). The lunisolar gravitational potential $\Omega_g$ is calculated by considering the real-time position of the Sun and Moon with an
ephemeris approach (Duffett-Smith, 1990). Except for the lunisolar gravitational potential term, two correction terms should
also be considered. The solid Earth tide $\Omega_e$ driven by the Sun and Moon has a counteracting effect to the ocean tides, which
is expressed as a portion of the lunisolar gravitational potential. Thus, $\Omega_g + \Omega_e$ can be written as $\alpha\Omega_g$, where $\alpha$ represents
the effective Earth elasticity factor. Though $\alpha$ is distinct for diurnal tidal components (Wahr, 1981), an identical factor of 0.69

is applied in this work following Kantha (1995). The self-attraction and loading (SAL) effect $\Omega_{SAL}$ considers the seafloor
deformation and the changing of the earth gravity field caused by the weight distribution of seawater. To determine the SAL
effect, one needs an explicit calculation based on a convolution integral (e.g., Shihora et al., 2022). However, this method is
costly, and thus we use a simplified form $\beta g\eta$. $\eta$ and $g$ represent sea surface elevation and the gravitational acceleration. The
scale factor $\beta$ has been estimated in previous work (e.g., Accad and Pekeris, 1978; Ray, 1998; Stepanov and Hughes, 2004)

and is set to 0.1 here.





### 2.3 The CVMIX_TIDAL parameterisation

The CVMIX_TIDAL parameterisation is implemented into FESOM2 along with the CVMIX library (Scholz et al., 2022). It is expressed as additional vertical diffusivity and vertical viscosity terms in the model. Take vertical diffusivity ($k_v$) as an example.

$$k_v' = k_v + \frac{\Gamma \epsilon}{N^2} = k_v + \frac{q \Gamma E(x,y) F(z)}{\rho N^2} \tag{2}$$

In Eq. (2), the term $k_v$ represents the original vertical diffusivity, which is the sum of constant background vertical diffusivity and KPP vertical diffusivity in this work. $\rho$ and $N$ denote seawater density and buoyancy frequency, respectively. The mixing efficiency $\Gamma$, which is set to 0.2 following Osborn (1980), represents the rate of dissipated mechanical energy that leads to oceanic mixing. $\epsilon$ represents the tidal energy dissipation rate, which can be further expanded as $qE(x,y)F(z)/\rho$. The tidal dissipation efficiency $q$ is estimated as 0.33, indicating that one-third of the tidal energy is dissipated locally, while the rest is radiated away and dissipated remotely as propagating low mode internal waves. Note that the remotely dissipated energy is already contained in the background vertical diffusivity in an OGCM (Simmons et al., 2004). The horizontal and vertical distribution functions of tidal energy dissipation are expressed as $E(x,y)$ and $F(z)$, which can be further expanded as below.

$$E(x,y) = \frac{1}{2} \rho_0 N_b \kappa h^2 \langle u^2 \rangle \tag{3}$$

$$F(z) = \frac{e^{-(H+z)/\zeta}}{\zeta(1 - e^{-H/\zeta})} \tag{4}$$

In Eq. (3), $E(x,y)$ means the parameterised energy conversion rate from barotropic to baroclinic tides. $\rho_0$ and $N_b$ represent the reference seawater density and bottom buoyancy frequency, respectively. $\langle u^2 \rangle$ is the temporal mean square tidal velocity, representing the barotropic tide energy. $\kappa$ and $h$ are the wavenumbers and amplitude scales of ocean topography, representing the bottom roughness. It should be noted here that $\kappa$ is a tuning variable so that the global integral of $E(x,y)$ fits the general value (Munk and Wunsch, 1998, 1 TW). In Eq. (4), the vertical distribution function $F(z)$ is a bottom intensified exponential profile whose depth integral is unity. $\zeta$, which is suggested as 500 m, denotes the e-folding scale of $F(z)$ from the bottom $-H$. The additional vertical viscosity is equal to the additional vertical diffusivity. That means the parameterisation considers the viscosity–diffusivity ratio (also known as the Prandtl number) as unity, which consists with the Prandtl number in the KPP scheme beneath the surface boundary layer.

### 2.4 Sensitivity runs

We set three experiments with the above configuration to investigate the influence of tide-induced mixing on the ocean state. The control run is named NOTIDE, which does not consider any kind of tidal effect. The first comparison run, LSTIDE, considers the lunisolar tidal potential in the momentum equations. The second comparison run, CVTIDE, does not simulate real ocean tides but applies the parameterisation of tide-induced mixing (CVMIX_TIDAL). The information of the sensitivity





runs is listed in Table 1. It should be noted that all other model configurations are identical among the three sensitivity runs. Following the spin-up strategy of Danek et al. (2019), the three sensitivity experiments start from the PHC3 climatology and are periodically forced by the CORE-II forcing. Each experiment is conducted for five consecutive cycles, namely, R1 to R5. Each cycle corresponds with the year 1948 to 2009. The spin-up strategy is demonstrated intuitively in Table 2. Unless otherwise

specified, all the model results shown in this paper are the average of the last 50 model years (1960 to 2009 in R5).

## 3 Validation of the tidal potential module

Though the tidal potential module has been applied in previous work (e.g., Müller et al., 2010; Weber and Thomas, 2017), the performance of this module in FESOM2 still needs to be evaluated. One crucial factor that should be carefully treated is the model timestep. Thus, a set of sensitivity runs is conducted to find the best timestep for the CORE-II mesh. The sensitivity

runs are configured with timesteps set to 300, 360, 400, 450, 600, 720, 900, 1200, and 1800 s, respectively. Each experiment is integrated for one year, and the hourly model output of sea surface height is applied for harmonic analysis using the T_TIDE package (Pawlowicz et al., 2002). Figure 1 shows the co-tidal charts of M2 tides from all sensitivity runs, demonstrating that the amplitudes of M2 tides are getting larger when we reduce the model timestep. However, the Greenwich phase lags are not significantly changed except for the last two runs (with timesteps set to 1200 and 1800 s). The reason is that the timesteps

of the last two runs are too long to resolve the propagation of tidal waves, leading to damped tidal amplitudes and distorted patterns of co-phase lines in, for example, the South Pacific Ocean.

To quantitatively determine the timestep for the best performance of the tidal potential module applying CORE-II mesh, we compare the harmonic tidal constants from the model results with those from tide gauge data. The University of Hawaii Sea Level Centre research quality data (Caldwell et al., 2015) is applied in this work. We obtain 687 tide gauge series at 540 tide

gauge stations. Note that some tide gauge series refer to the same station and we integrate those into one. We exclude some tide gauge stations because: (1) these stations are not located in the CORE-II mesh; (2) the water depths at these stations in the model are shallower than 200 m. We exclude shelf tide gauges because global tide models do not have a good performance in shelf areas inherently. Improving model results in shelf areas may lead to a larger error in the global view. A similar approach is also used in previous work that compares model results with pelagic tide gauges (Müller et al., 2010). Since most tide gauges

span over multiple years, each tide gauge series is truncated into several yearly segments according to the calendar year. For the preciseness of analysis, segments with a completeness index (proportion of data span without missing data) lower than 0.8 are discarded. Tidal harmonic analysis is done to each annual segment at each station, and the harmonic tidal constants at each station are thus averaged from multiyear constants. Finally, 229 tide gauge stations are applied for the comparison with model results.

A root-mean-square error (RMSE), referring to Cummins and Oey (1997), is defined as below to estimate the deviations of model results.

$$rmse = \sqrt{\frac{1}{N} \sum_{i=1}^{N} \left[ \frac{1}{2} \left( A_o^2 + A_m^2 \right) - A_o A_m \cos \left( \phi_o - \phi_m \right) \right]} \qquad (5)$$





In Eq. (5), $A_o$ and $\phi_o$ represent tidal amplitudes and Greenwich phase lags derived from the observational data, while $A_m$ and $\phi_m$ represent those derived from the model results. It should be noted here that the RMSE defined in Eq. (5) can be

evaluated either for one tidal component throughout all tide gauges or several tidal components at one tide gauge. In the first case, $N = 229$ for one tidal component throughout all 229 tide gauge stations (see Fig. 2a); in the second case, $N = 2$ for two main diurnal/semidiurnal tidal components at one tide gauge station (see Figs. 2b and 2c). Fig. 2a shows the RMSEs at all tide gauge stations for each sensitivity run's four main tidal constituents. It can be found that the last two sensitivity runs present significant errors for M2 tides. As we claim before in Fig. 1, a large timestep may not resolve fast-travelling barotropic tides,

leading to the damping of tide amplitudes. By considering all RMSEs of the four main tidal constituents, one can find that an optimal timestep lies between 450 s and 720 s. Considering the computational cost, we finally choose a timestep of 720 s to simulate global tides with the CORE-II mesh. With the timestep set to 720 s, the RMSEs for diurnal (O1 and K1) and semidiurnal (M2 and S2) tidal components at each tide gauge station is further shown in Figs. 2b and 2c, respectively. It can be seen that semidiurnal tides have larger deviations than diurnal ones, and gauges close to the coasts have larger deviations than

those in basin centres.

In general, the tidal potential model demonstrates good consistency with tide gauges in most areas. However, we should state that some tide gauges also show considerable inconsistency from series to series and from year to year. For example, the Greenwich phase lags at some stations may have a discrepancy exceeding 120° among different years. That would introduce significant errors in estimating harmonic tidal constants at these stations. We speculate that the inconsistency may come from

the changing of observational instruments.

## 4 Model results

### 4.1 Hydrography

We compare model results with the World Ocean Atlas 2018 (Locarnini et al., 2018; Zweng et al., 2018, WOA18). The decadal-averaged annual dataset with a resolution of 0.25° is obtained. For comparability, the in situ temperatures from the

WOA18 dataset are converted to potential temperatures. Unless otherwise specified, temperatures are referred to as potential temperatures in the below context.

Figure 3 shows the temperature biases between the sensitivity runs and WOA18. In the upper layer, LSTIDE generally reduces the temperature biases in the low-latitude areas of the three major oceans, indicating lower temperature biases than CVTIDE. However, in the subpolar North Pacific Ocean, CVTIDE shows a better reduction of the warm biases than LSTIDE. In

the intermediate layer, LSTIDE generally reduces the temperature biases in most mid- and low-latitude areas. But CVTIDE has lower temperature biases in the whole Arctic region. In addition, neither CVTIDE nor LSTIDE presents major improvements to the Southern Ocean. In the deep layer, CVTIDE shows a cooling effect in the Southern Ocean, but LSTIDE shows a warming effect in the basins of the three major oceans, showing lower temperature biases in the global view. Figure 4 is the same as Fig. 3 but for salinity. In the upper layer, CVTIDE mainly reduces the salinity biases in the tropical Pacific Ocean. In contrast,

LSTIDE reduces the salinity biases in the tropical Pacific Ocean and the tropical Indian Ocean. In the intermediate layer,



CVTIDE reduces salinity biases in the tropical Pacific Ocean and the tropical Indian Ocean. In contrast, LSTIDE reduces salinity biases in the North Pacific Ocean and North Atlantic subtropical gyre area. In the deep layer, only LSTIDE shows improvements in the low-latitude area in the Atlantic Ocean.

The tidal effects also demonstrate different degrees of improvements at different depths of each basin. Figure 5 shows basin averaged temperature and salinity biases at each vertical layer in the model. In the left panels, LSTIDE shows significant improvements to the temperature biases in the Pacific, Atlantic, and Indian Oceans deeper than 1000 m. CVTIDE only reduces temperature biases in the Arctic Ocean at 500–2000 m in depth. Also, both CVTIDE and LSTIDE reduce temperature biases in the Southern Ocean deeper than 4000 m. In the right panels, improvements can only be found in LSTIDE. Salinity biases are slightly reduced in the Atlantic and Pacific Oceans, around 1500 m in depth. It should also be noted that in Fig. 5j, large salinity biases emerge in the surface Arctic Ocean. That comes from the differences between the PHC3 and WOA datasets. Our model starts with the PHC3 climatology, which integrates the WOA dataset and the Arctic Ocean Atlas.

Both CVTIDE and LSTIDE also greatly influence the hydrography in continental shelf and slope areas, which cannot be clearly shown in Figs. 3–5. By assembling grids with the same bathymetry and averaging the temperature and salinity biases, the bathymetry averaged biases with respect to depths are shown in Fig. 6. In Fig. 6a, the model results in the control run demonstrate significant temperature biases near continental shelf and slope topographies, and both CVTIDE and LSTIDE decrease the biases in these areas. By comparing Fig. 6c with Fig. 6e, CVTIDE has a better improvement near the bottom of continental shelf and slope, while LSTIDE has a significant reduction of temperature biases at 1000–4000 m in depth, which is not shown in CVTIDE. In Fig. 6b, significant salinity biases in the control run concentrate in the shallow continental shelf areas. It is shown in Figs. 6d and 6f that CVTIDE decreases salinity biases at near-bottom continental shelf area, while LSTIDE decreases salinity biases in the whole layer of continental shelf area. Neither of the two comparison runs demonstrates a significant improvement to other areas, except for the slight improvement of LSTIDE around 1500 m in depth.

## 4.2 Sea ice

Compared with the control run, both tidal effects alter the sea ice results, but the patterns are different. Figure 7 shows the results of sea ice thickness (SIT) in the Arctic region. In the Arctic region, CVTIDE generally shows SIT decrease. In March, a remarkable SIT decrease occurs in the Labrador Shelf (along with the Hudson Strait, the Davis Strait, and the Baffin Bay) and the Okhotsk Sea. In September, an SIT decrease occurs in the Baffin Bay, the northeast Greenland Shelf, and the north Barents Shelf. But an SIT increase emerges in the Canadian Arctic Archipelago (CAA) in CVTIDE. LSTIDE also shows an SIT decrease in the Okhotsk Sea and Labrador Shelf but is weaker than CVTIDE. Unlike CVTIDE, LSTIDE shows a remarkable SIT decrease in the CAA. The mean SIT in the CAA in the control run is 6.01 m (5.23 m) in March (September). LSTIDE decreases 0.60 m (0.67 m), while CVTIDE increases 0.03 m (0.03 m). In addition, SIT slightly increases in the Greenland–Iceland Sea (March), the Baffin Bay, the northeast Greenland Shelf, and the north Barents Shelf (September) in LSTIDE, and that is also different from CVTIDE.

Figure 8 is the same as Fig. 7 but for Antarctica. In March, both tidal effects show similar changes in SIT around Antarctica, but CVTIDE is more prominent than LSTIDE. In September, CVTIDE and LSTIDE also show similar patterns, including a


general SIT decrease around Antarctica and an SIT increase on the Weddell Sea Shelf and the Ross Sea Shelf. But in the Western Antarctica Shelf, SIT decreases in CVTIDE while increases in LSTIDE. In addition, in the area ranging from 30° E to 60° E, SIT increases in CVTIDE but decreases in LSTIDE.

It should also be pointed out that in both Fig. 7 and Fig. 8, CVTIDE consistently reduces SIT at shelf-break areas, which are indicated by the grey dashed lines. However, LSTIDE is not showing this consistency. The difference is further discussed
in Sect. 5.

### 4.3  Mixed layer depth

Two definitions of mixed layer depth (MLD) are computed in FESOM2. The first one is defined as the shallowest depth where the vertical buoyancy gradient is equal to a local critical buoyancy gradient, following Large et al. (1997). The results are shown in Fig. 9. In the North Pacific Ocean, CVTIDE deepens MLD in the Sea of Okhotsk and the Bering Sea, while LSTIDE shows
significant changes in the Kuroshio–Oyashio front. The pattern features a "blue-to-red" dipole, indicating an equatorward displacement of the deep mixed layer area near the Kuroshio. In the North Atlantic Ocean, LSTIDE shows a deepening effect on the MLD, while CVTIDE shows a shoaling effect on the MLD in the centre Labrador Sea. In the Southern Ocean, both CVTIDE and LSTIDE deepen MLD in the area ranging from 180° to 60° W, but the effect of CVTIDE is stronger than that of LSTIDE.

The second kind of MLD, following Monterey and Levitus (1997), is defined as the depth where the density over depth differs by 0.125 sigma units from the surface density. This definition shows larger MLDs in the deep convection area compared with the previous one. As shown in Fig. 10, the MLDs in the Labrador Sea and the Weddell Sea can be greater than 3000 m, and both CVTIDE and LSTIDE deepen the MLDs in the deep convection areas. In the Labrador Sea, the deepening effect of LSTIDE (500 m) is stronger than that of CVTIDE (200 m). In the Weddell Sea, both CVTIDE and LSTIDE extend regions
with deep mixed layers to the east (ranging from 30° W to 0°), but the deepening effect of CVTIDE (1700 m) is much stronger than that of LSTIDE (460 m).

### 4.4  Meridional Overturning Circulation

Both CVTIDE and LSTIDE influence the MOC. As shown in Fig. 11c and 11e, CVTIDE and LSTIDE show different effects in the AMOC. CVTIDE weakens the AMOC upper cell by 0.50 Sv and strengthens the lower cell by 0.75 Sv, while LSTIDE
strengthens the AMOC upper cell by 1.50 Sv and weakens the lower cell by 0.25 Sv. That indicates CVTIDE weakens the North Atlantic Deep Water (NADW) formation and strengthens the Antarctic Bottom Water (AABW) formation, while LSTIDE shows the opposite.

The results of PMOC are shown in the right column of Fig. 11. The main effect of CVTIDE on the PMOC occurs in the south Indo-Pacific Ocean. The deep water upwelling at 20° S is enhanced for 4 Sv. The LSTIDE has a much more significant
effect than CVTIDE. Figure 11f features two cells: the south cell upwells deep water in the tropical Indo-Pacific Ocean, while the north cell upwells deep water in the subarctic North Pacific Ocean. Both cells enhance deep water upwelling for 5 Sv, and the upwelled deep waters are released to the intermediate layer (about 1000–2000 m) of the Indo-Pacific Ocean. It should also





be noted here that the north cell in Fig. 11f shows consistency with the North Pacific bathymetry, which is further discussed in Sect. 5.

### 4.5 Vertical diffusivity

The results of vertical diffusivity in the three sensitivity runs are shown in Fig. 12. The middle column shows that CVMIX_TIDAL parameterisation mainly enhances vertical diffusivity in the deep ocean. The difference between CVTIDE and NOTIDE is most significant at the bottom layer and is dependent on the bottom topography roughness. Vertical diffusivity is enhanced in seamounts, deep ocean ridges, and shelf-break areas. The pattern of LSTIDE is much different compared with CVTIDE, and we cannot see a coherent increase of vertical diffusivity in the deep/bottom layer ocean. We should also notice that the differences between CVTIDE/LSTIDE and NOTIDE in the upper and intermediate layers show similar patterns with the MLD results (Fig. 9). That indicates deep mixed layers are related to high vertical diffusivities in the upper ocean.

### 4.6 Barotropic streamfunction

The barotropic streamfunctions in the three sensitivity runs are shown in Fig. 13. CVTIDE and LSTIDE show similar patterns in changing the global barotropic streamfunction. The patterns of streamfunction differences in Figs. 13b and 13c mainly consist of two loops. The "blue loop" is mainly located in the Indo-Pacific Ocean: the southward transport enhances the ITF and finally contributes to the ACC, while its recirculation locates in the western Pacific Ocean. The "red loop" enhances the ACC separately. The blue and red loops are stronger in LSTIDE than those in CVTIDE. CVTIDE enhances the ITF and the ACC through the Drake Passage by 1.72 Sv and 4.79 Sv, while those values for LSTIDE are 3.92 Sv and 10.34 Sv, respectively. In addition, we can find an important difference regarding the blue loop: the blue loop extends to the subpolar Pacific Ocean and the South Atlantic Ocean in LSTIDE but not in CVTIDE. The difference determines the PMOC–AMOC connection, which is mentioned again in Sect. 5.

### 4.7 Energy

Several energy terms are diagnosed from the model results, and the global integration values of these energy terms are shown in Table 3. Due to the additional energy input in LSTIDE (see barotropic tide power and kinetic energy), two energy dissipation terms (bottom drag and viscous dissipation) are enhanced compared with NOTIDE and CVTIDE, especially for bottom drag. In addition, both CVTIDE and LSTIDE enhance buoyancy flux compared with NOTIDE. The horizontal distributions of viscous dissipation, bottom drag, and buoyancy flux in the three sensitivity runs are further shown in Fig. 14. From Fig. 14c, we can see that LSTIDE enhances bottom drag significantly. In most areas, the enhancement is about one order, but it can reach an order of 5 in several marginal seas because of the strong tidal currents. Because of the addition of explicit tidal velocities, the bottom drag and the vertical shear of horizontal velocity are enhanced, especially on continental shelf areas. Thus, LSTIDE also enhances viscous dissipation, which is related to vertical shear. It should be noted here that the vertical viscosity is also enhanced in the CVMIX_TIDAL parameterisation, but the effect is negligible (see Fig. 14d). From the bottom row of Fig. 14,





we can find that compared with CVTIDE, LSTIDE has a stronger enhancement of buoyancy flux in the Kuril–Aleutian Ridge

and Indonesian Archipelago but weaker in other mid-latitude areas.

## 5   Discussion

### 5.1   The effect of resolution on tidal mixing

As shown in Munk and Wunsch (1998, their Fig. 4), internal tides are important linking between surface tides and oceanic
mixing. Indeed, Niwa and Hibiya (2011, 2014) demonstrate that the conversion rate of internal tide energy in a global tide
model depends on the model resolution. Thus, if the model mesh cannot resolve internal tides, the internal-tide-induced mixing
is missing in the model result. Figure 15 presents the ratio of mode-1 internal tide wavelengths to the mesh resolution. We can
find that most of the model areas cannot resolve propagating internal tides in the mid-latitude area, especially for semidiurnal
ones in Fig. 15b. In addition, even though the unstructured model mesh has higher resolution in shelf-slope areas, it still cannot
resolve internal tides therein because the wavelengths of internal tides are smaller in these areas. So we infer that LSTIDE has
a strong dependence on the model resolution.

Evidence can be found in the model results. In Fig. 6, compared with CVTIDE (Figs. 6c and 6d), LSTIDE (Figs. 6e and
6f) shows a weaker reduction of hydrographic biases near the bottom of shelf breaks. The internal tides at shelf-slope areas
cannot be resolved in LSTIDE, which weakens the mixing effect. Tidal mixing on shelf breaks also affects SIT. Unlike other
areas, where the temperature decreases monotonically with depth, the polar regions have colder water near the surface than
in the deeper layers. Thus in polar areas, the tidal mixing on shelf-slope areas would lead to a warmer surface ocean, which
should decrease the SIT. In our result, LSTIDE shows a smaller SIT decrease than CVTIDE on the shelf-slope area, such as
the Labrador Shelf in Figs. 7b and 7c. Similar results can also be found near the West Antarctica Shelf (Figs. 8e and 8f). The
effect of the resolution is also reflected in the model vertical diffusivity. Both observational results (Polzin et al., 1997) and
CVMIX_TIDAL parameterisation suggest larger vertical diffusivities near the bottom, which is caused by the tide–topography
interaction. However, vertical diffusivities in the deep and bottom layers are not generally higher in LSTIDE than in NOTIDE
(Figs. 12i and 12l). The lack of resolving internal tides causes the underestimation of vertical shear, thus leading to a different
result compared with CVTIDE. The buoyancy flux results (Figs. 14h and 14i), which are most directly connected to oceanic
upwelling, also show less mixing effects in mid-latitude areas in LSTIDE compared with CVTIDE. Figure 15 shows the model
mesh can roughly resolve propagating internal tides in tropical areas, where strong mixing can be found in LSTIDE. Figure
11f shows strong mixing and upwelling in the equatorial Indo-Pacific Ocean, while this is not clearly shown in CVTIDE (Fig.
11d). We think that this comes from the underestimation of tidal dissipation efficiency ($q$). The tidal dissipation efficiency is set
to 0.33 in this work, but previous work indicates higher efficiencies in the Indonesian Archipelago area (Koch-Larrouy et al.,
2007; Nagai and Hibiya, 2015).

Figure 15 also implies that there are critical latitudes for propagating internal tides. According to the dispersion relationship
of internal tides, where the inertial frequency is higher than a tidal frequency ($f > \omega$), internal tides at the tidal frequency
would not have a wave solution, resulting in trapped internal tides instead of propagating internal tides (light grey areas in Fig.





15). Unlike propagating internal tides, trapped internal tides can only be dissipated in a small domain, leading to a higher local dissipation rate and thus stronger local mixing. The Kuril Ridge and Aleutian Ridge, which connect the Sea of Okhotsk and the Bering Sea with the North Pacific Ocean, are the main generation sites for trapped diurnal internal tides. Large dissipation
rates by trapped internal tides are validated in the previous modelling research (Niwa and Hibiya, 2011, 2014; Tanaka et al., 2010, 2013; Falahat and Nycander, 2015), and intense mixing is also observed therein (Nakamura et al., 2010; Itoh et al., 2014).

  LSTIDE demonstrates intense mixing in the Kuril Ridge and Aleutian Ridge, resulting in higher vertical diffusivity and buoyancy flux than NOTIDE in these areas (Figs. 12l and 14i). The strong mixing upwells an additional 5 Sv of deep water
in the North Pacific (around 50° N) to the intermediate layer and then spreads to the south (Fig. 11f). The intermediate water then contributes to the ITF (Fig. 13c) and flows into the Indian Ocean. Stronger PMOC helps to reduce temperature biases in the intermediate Indo-Pacific Ocean (Fig. 3f): the upwelling of the deep water reduces the warm biases in the North Pacific Ocean and cold biases in the tropical Indo-Pacific Ocean in NOTIDE. The formation of intermediate water also pushes the Kuroshio–Oyashio front to the south, and that explains why we have "blue-to-red" dipole patterns for MLD (Fig. 9c) and
vertical diffusivity (Fig. 12c) in the North Pacific.

  Trapped internal tides and related mixing can be simulated with LSTIDE, because the CORE-II mesh has higher resolution in the high-latitude areas and the trapped internal tides are not forming as travelling waves. As a comparison, even though CVTIDE does not show much influence on the PMOC in the North Pacific Ocean, it shows similar but weaker patterns in reducing the temperature biases in the intermediate layer. This probably comes from two reasons: (1) most of the trapped
internal tides' energy is dissipated locally, so the tidal dissipation efficiency is close to unity ($q \approx 1$) in this area instead of 0.33 from CVTIDE; (2) the simulation is not long enough to get stable. It should also be emphasised here that even though the PMOC lower cell in the North Pacific is reported in most previous work (e.g., Simmons et al., 2004; Oka and Niwa, 2013), our results in LSTIDE demonstrate a different shape for the PMOC lower cell in the North Pacific. The significant upwelling occurs in the Aleutian Ridge, and the upwelling has great consistency with the topography (take the brown transection in Fig. 11f as
an example). It seems that the diurnal tidal mixing beyond critical latitudes causes an upwelling along the bottom boundary, which is not reported in previous work. Nevertheless, our result supports some previous arguments (Scott and Marotzke, 2002; Ferrari et al., 2016) that the upwelling caused by near boundary mixing is more important than interior mixing in closing the THC.

## 5.2 The tidal effects on the global THC

The model also shows interesting results in the spin-up. The time series of several main constituents of the global THC is demonstrated in Fig. 16. In Figs. 16a and 16b, we find that both CVTIDE and LSTIDE demonstrate a stronger PMOC and ITF than the control run since the beginning. In the fourth and fifth cycles, the discrepancies of PMOC and ITF in strength between CVTIDE/LSTIDE and NOTIDE become stable. In Fig. 16c, it is shown that in the first cycle, the strength of AMOC upper cell in both CVTIDE and LSTIDE is weaker than that in the control run. However, since the second cycle, the strength
of the AMOC upper cell in LSTIDE starts to surpass that in NOTIDE. The strength of the AMOC upper cell in CVTIDE is





weaker than that in NOTIDE in all cycles. As mentioned in Sect. 2, the CVMIX_TIDAL scheme applies the same formula to the model viscosities, potentially increasing model dissipation and leading to a weaker circulation on shelf-slope areas. Thus, the wind-driven circulation, the poleward heat flux and the AMOC upper cell are weakened, which corresponds with previous work (e.g., Jayne, 2009; Yu et al., 2016). But a major difference between CVTIDE and LSTIDE embodies the subsequent

enhancement of the AMOC upper cell in LSTIDE. Results from LSTIDE reveal that the upwelling in the North Pacific and Indonesian Archipelago links to the North Atlantic. Figures 11f and 13c reveal that the enhancement of mixing in the North Pacific and Indonesian Archipelago results in a stronger PMOC and ITF. The ITF mass transport finally forms the "blue loop" in the barotropic streamfunction difference (Fig. 13c). The westernmost part of this loop extends to the South Atlantic, which seems to have no connection with the North Atlantic. But note here the barotropic streamfunction has an offset effect in the

vertical direction. Figure 11e further shows that the upper cell of AMOC is enhanced and the effect originates from the South Atlantic. On the contrary, even though CVTIDE also has a stronger ITF and a "blue loop" streamfunction difference, it does not strengthen the upper cell of AMOC because the "blue loop" is restricted in the Indo-Pacific Ocean and does not extend to the South Atlantic (Fig. 13b). The cycle revealed in our result corresponds well with the schematic diagram in (Talley et al., 2011, her Fig. S14.1c). The enhancement of ITF due to tidal mixing is also reported in previous research (Schiller, 2004; Sasaki

et al., 2018), and the link between stronger PMOC lower cell and stronger AMOC upper cell can also be found in Melet et al. (2016).

As to the AMOC lower cell, LSTIDE shows a weakening effect while CVTIDE shows a strengthening effect (Fig. 16d). It is widely believed that the global tidal mixing and wind-driven upwelling in the Southern Ocean affect the MOC (Ledwell et al., 2000; Marshall and Speer, 2012). Since the model configuration applies the same atmospheric forcing in the three

sensitivity runs, the reason concerning wind stress would be excluded. Moreover, considering the vertical advection–diffusion balance (Munk and Wunsch, 1998; Prange et al., 2003), the tidal mixing should strengthen the MOC, as CVTIDE shows. As discussed above, LSTIDE has the disadvantage in demonstrating tide-induced mixing in the vast mid-latitude areas due to the resolution; thus, the AMOC lower cell is weakened instead of strengthened in LSTIDE. We also guess that the enhanced vertical diffusivities may be added to deep convection areas, directly leading to stronger convection. However, the CVMIX_TIDAL

parameterisation implies that the enhanced vertical diffusivities are not added to areas where gravitational instability occurs ($N^2 < 0$). So we finally conclude that the AMOC lower cell is enhanced due to tidal mixing, and more AABW formation in the Weddell Sea is a result instead of a cause of the AMOC lower cell strengthening. A similar conclusion can also be found in Exarchou et al. (2012).

CVTIDE and LSTIDE also demonstrate that the ACC transport through the Drake Passage is larger than that in NOTIDE

(Figs. 13 and 16e). We can see from Fig. 16e that the strength of ACC in all three sensitivity runs decreases during the five cycles, but the strength decrease more slowly in the comparison runs, which can be explained with Fig. 17. Figures 17d and 17f show that both comparison runs have similar effects on changing potential density in the southern hemisphere. In the mid- and low- latitude areas, both tidal effects demonstrate denser seawater in the upper ocean and lighter seawater in the lower ocean, indicating the promotion of ocean potential energy caused by the tidal mixing. In the high-latitude areas, both comparison runs

show denser seawater from surface to bottom, indicating the reduction of ocean potential energy due to the deep convection.





In depths below 1000 m, both CVTIDE and LSTIDE show potential density decrease at subtropical latitudes. The decrease of potential density is caused by the mixing and advection effect indicated by the global MOC (GMOC). For example, Fig. 17e shows that strong mixing and upwelling take place in the tropical region, and the upwelled water is further advected to the south (similar to Fig. 11f). The advection effect causes the "blue tongue" pattern in Fig. 17f (1000–2000 m in depth). In

addition, the increase of potential density at subpolar latitudes can be explained by the AABW formation. In the above context, our results show that the tidal mixing in CVTIDE causes the strengthening of the AABW formation. This is also why the potential density increase at subpolar latitudes is more substantial in CVTIDE than that in LSTIDE. Even though CVTIDE and LSTIDE alter potential density differently, both experiments increase the meridional density gradients in the Southern Ocean, which prevents the strength of ACC from decreasing due to the thermal wind relation. Thus we conclude that both CVTIDE

and LSTIDE affect the ACC strength by enhancing the meridional density gradient. Similar results are also shown in Saenko and Merryfield (2005) that with tidal mixing parameterisation, the ACC transport is stronger by 25% compared with a no-tide run.

### 5.3   Other tidal effects: what does CVTIDE miss?

The "barotropic tide–baroclinic tide–oceanic mixing" chain is considered the most crucial tidal effect on the ocean and climate.

But we are still questioning if there are any effects from tides other than internal tide mixing. Since CVTIDE only considers internal tide mixing effects and LSTIDE considers real tide effects, we can answer this question by comparing LSTIDE with CVTIDE.

    The most significant difference is the energy in the model. If the tidal potential term is added to the ocean primitive equation, tide energy is thus input to the system (Table 3). The global barotropic tide power is estimated as 4.7 TW, which is larger

than the reference values in previous research (e.g., Munk and Wunsch, 1998, 3.5 TW). We speculate the overestimation of barotropic tide power may come from the lack of internal tide simulation and thus internal wave drag (e.g., Jayne and St. Laurent, 2001; Sulzbach et al., 2021). External energy input not only leads to higher kinetic energy but also higher energy dissipation. Comparing Fig. 14b with Fig. 14c, we find that the bottom drag is significantly enhanced in LSTIDE than in CVTIDE. Regarding the viscous dissipation, LSTIDE shows remarkable enhancement on continental shelves (Fig. 14f), while

CVTIDE does not show significant changes compared with NOTIDE. The reason is that LSTIDE has strong tidal shear caused by bottom drag on continental shelves, while CVTIDE only increases viscosity and diffusivity on shelf breaks and seamounts, indicating an effect of internal tides.

    Additionally, we also find that considering the strong tidal shear on continental shelves can reduce model biases. Both Fig. 6e and Fig. 6f indicate that in LSTIDE, the temperature and salinity biases on continental shelves (e.g., bottom depth less

than 100 m) are better reduced compared with CVTIDE. Via a coupled climate model, Lee et al. (2006) also found that local hydrographic biases can be reduced when considering tidal shear effects on continental shelves.

    From a mathematical point of view, the addition of periodic tidal motions can also affect stress terms, leading to a tidal residual effect. Even though tidal velocities can roughly be eliminated via time-averaging, the stress terms can always leave none-zero cross terms. These effects are more remarkable in regional simulations than global ones. Our results show that only





the sea ice distribution in the CAA is significantly affected by the tidal residual effect. Figure 7 shows that in LSTIDE, the SIT in the CAA is reduced by about 10%. Adding strong tidal currents in the narrow channels can lead to an additional ocean-to-ice stress, and the theoretical basis can be referred to in Hibler (1979, his Eq. 3). Our results also show that the additional stress causes sea ice transports from CAA to the adjacent areas, such as the north Baffin Bay (Fig. 7f).

## 6 Conclusions

The crucial effect of tides on the ocean and climate has been widely discussed in previous work (e.g., Polzin et al., 1997; Munk and Wunsch, 1998; Ledwell et al., 2000). Moreover, it is generally implemented in an OGCM via two approaches: parameterised tidal mixing and explicit tidal forcing. In this work, we implement both modules in FESOM2 to assess the two methods in the same framework. By analysing and summarising different aspects from the model results, we draw the conclusions below.

First, the resolution of the CORE-II mesh is not fine enough to simulate internal tides and internal tide mixing. Thus, the real tide simulation (LSTIDE) is not as good as tidal mixing parameterisation (CVTIDE). For instance, our control run shows large temperature biases in near-bottom shelf-slope areas, which can be reduced by tidal mixing. Also, tidal mixing in polar regions can lead to lower SIT in the model because of the polar stratificational feature. Additionally, tidal mixing in the deep ocean can lead to a stronger AMOC lower cell, with stronger AABW formation and deeper MLD in the Weddell Sea. Nevertheless,

these phenomena are not significant in LSTIDE than in CVTIDE because LSTIDE generally underestimates the strength of tidal mixing.

Second, previous research has pointed out that applying 0.33 to the mixing efficiency ($q$) may cause underestimating tidal mixing in some areas. Our results further show that tidal mixing in the Indonesian Archipelago and Kuril–Aleutian Ridge are stronger in LSTIDE than CVTIDE. That is not contradictory with the above statement, because the resolution in these areas is

relatively high and trapped internal tides are different from propagating internal tides. Our results show that the tidal mixing in these two areas drives stronger upwelling in the deep Pacific Ocean. In the Kuril–Aleutian Ridge, deep water is upwelled to the intermediate layer and advected to the tropical region; in the Indonesian Archipelago, the upwelled deep water enhances the ITF along with the intermediate water from the north. The deep water upwelling in the Pacific Ocean and the ITF enhancement result in lower hydrographic biases in the intermediate and deep layers of the North Pacific and tropical Indo-Pacific Ocean,

which is more observably in LSTIDE than CVTIDE. Beyond that, a link between stronger PMOC upwelling and stronger AMOC upper cell is only found in LSTIDE.

Third, both tidal effects enhance the strength of the ACC. We conclude that this is caused by thermal wind balance: both CVTIDE and LSTIDE increase meridional density gradient in the Southern Ocean. However, the reason for the increasing of meridional density gradient is different: CVTIDE increases seawater density in the deep layers of subpolar regions due to

stronger AABW formation; LSTIDE decreases seawater density in the intermediate layers because of the PMOC difference.

Finally, we summarise some other tidal effects which cannot be presented in the CVMIX_TIDAL parameterisation. Our results show that when we implement real tide motion into the model, both total ocean kinetic energy and energy dissipation





increase. The globally integrated bottom drag is remarkably enhanced by almost one order. Real tides also lead to stronger viscous dissipation on continental shelves, indicating strong tidal shear and mixing in these shelf areas. Our results further

prove that the hydrographic biases in shelf areas are less reduced in CVTIDE than LSTIDE because the mixing process in shelf areas is not considered in the CVMIX_TIDAL parameterisation. In addition, the tidal residual effect is also significant in ocean-to-ice stress, which is verified in our results that the SIT in the CAA is reduced by about 10% and the sea ice is transported to the adjacent areas.

In conclusion, our work shows that both spatial and temporal resolution needs to be refined to simulate real tides in climate

research. Previous research indicates that a model usually requires at least $0.1°$ to carry out a global internal tide simulation (e.g., Shriver et al., 2012; Li et al., 2015), but no research conduct a run longer than 10 years with a resolution as high as $0.1°$ due to the enormous computational cost. Thus, for long-term climate study (e.g., Shi and Lohmann, 2016; Ackermann et al., 2020; Lohmann et al., 2020), it is more applicable to use a parameterisation scheme for climate simulations. Nevertheless, the real tide simulation also shows some phenomena that are not well expressed by the CVMIX_TIDAL parameterisation. For

example, the underestimation of the mixing efficiency parameter in the Indonesian Archipelago and Kuril–Aleutian Ridge, the linking between stronger PMOC upwelling and stronger AMOC upper cell, and the tidal effects on coastal and shelf areas are revealed in our results. The tidal effects, including mixing effects, are revealed from our ocean-only model results, but it should also be noted that the effects may be different in a coupled climate model. In addition, vertical mixing might have changed over Earth history and provide a possible mechanism for past climate changes and climate latitudinal gradients (Lohmann, 2020).

Thus, it is important to consider tidal effects in climate research. Our future work on climate research considering tidal effects is to modify a tidal parameterisation scheme via a high-resolution short-term real-tide case, as well as to evaluate tidal mixing schemes from other theoretical frameworks (e.g., Olbers and Eden, 2013).

*Code and data availability.* The FESOM2.1 source code applied in this work is available from https://github.com/FESOM/fesom2/releases/tag/2.1.0 under the GPL-3.0 licence (last access: 1 August 2021). The last 50 years averaged model results, as well as the postprocessing and visu-

alisation codes, are archived on Zenodo (Song et al., 2021). The whole simulation results can be obtained by contacting the corresponding author. The tide gauge data can be downloaded from the University of Hawaii Sea Level Centre's website (last access: 27 July 2020). The PHC3 climatology, WOA18 climatology and CORE-II forcing are freely available online.

*Author contributions.* P. Song and D. Sidorenko implemented the tidal module into FESOM2. P. Scholz implemented the CVMIX package into FESOM2. P. Song performed all the experiments. All authors contributed to the analysis of the results.

*Competing interests.* The authors declare that they have no conflict of interest.



*Acknowledgements.* The authors thank Nikolay Koldunov for the help on FESOM2 settings and Sergey Danilov for discussing the model results. P. Song is funded by the ***China Scholarship Council***. G. Lohmann and M. Thomas received funding through the topic ***Ocean and Cryosphere under climate change*** in the Program ***Changing Earth - Sustaining our Future***. The experiments were performed at the AWI Computing Centre.





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

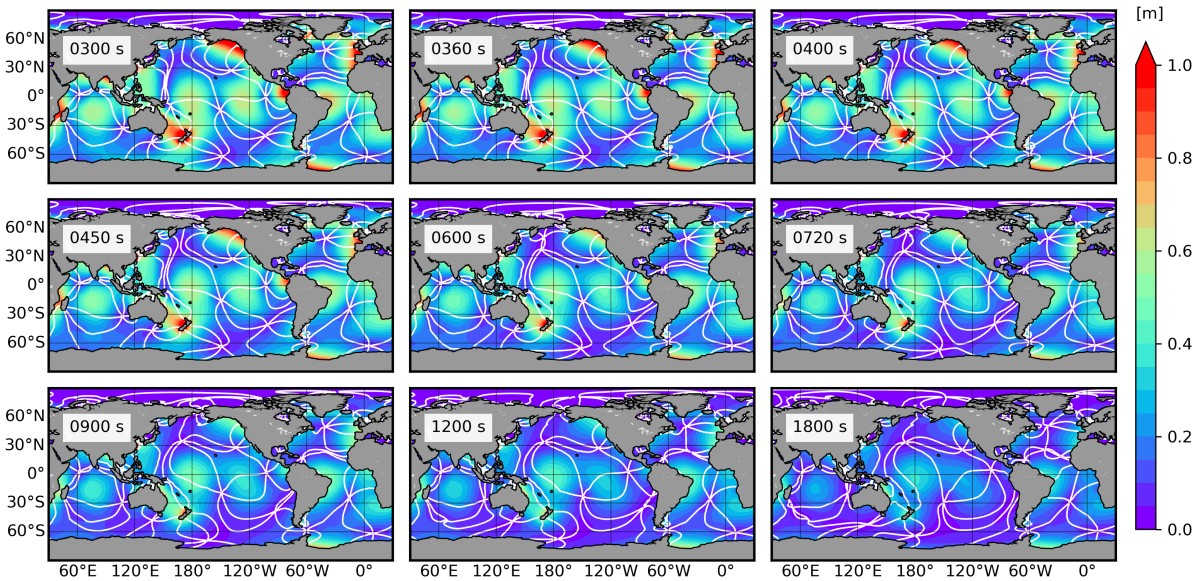

**Figure 1.** Co-tidal charts of M2 tides from the timestep sensitivity runs. The colours indicate tide amplitudes, and the white lines indicate Greenwich phase lags with an interval of $60°$. The timestep setting in each run is labelled on each panel.





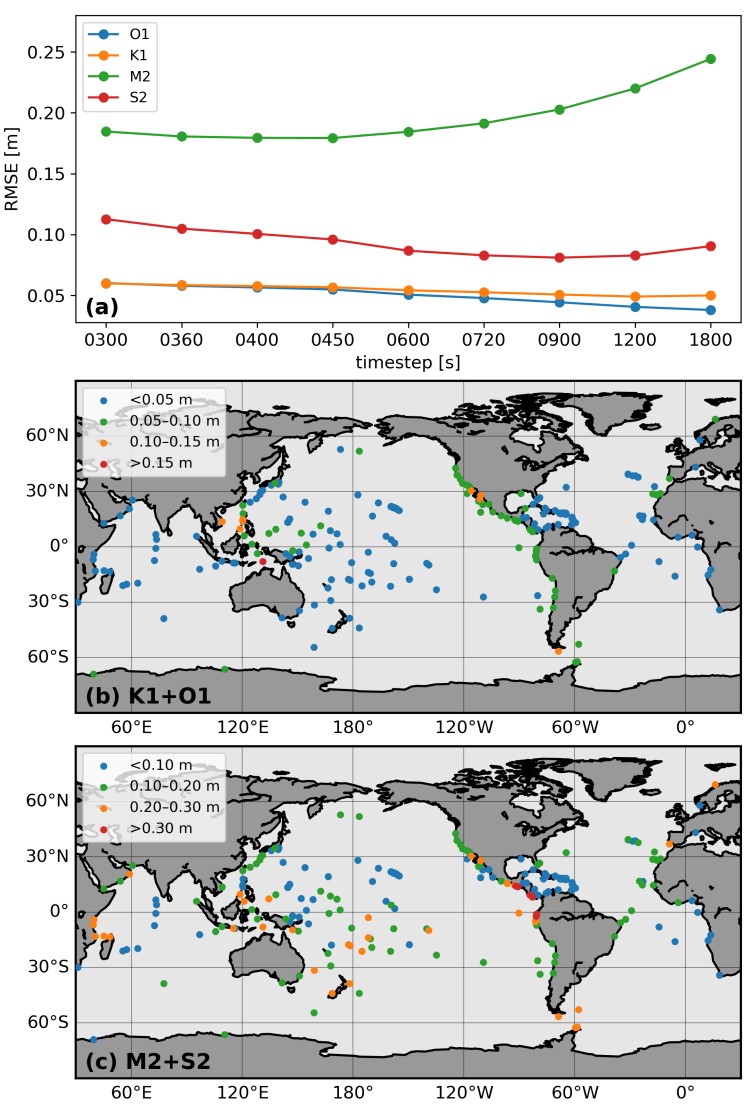

**Figure 2.** (a) RMSE of the four main tidal components (O1, K1, M2, and S2 tides) for all 229 tide gauges; (b) In the sensitivity run with timestep set to 720 s, RMSE of main diurnal tidal components (O1 and K1 tides) at each tide-gauging station. Scatter points are classified into four levels by the magnitudes of RMSE; (c) The same as panel (b) but for main semidiurnal tidal components (M2 and S2 tides).

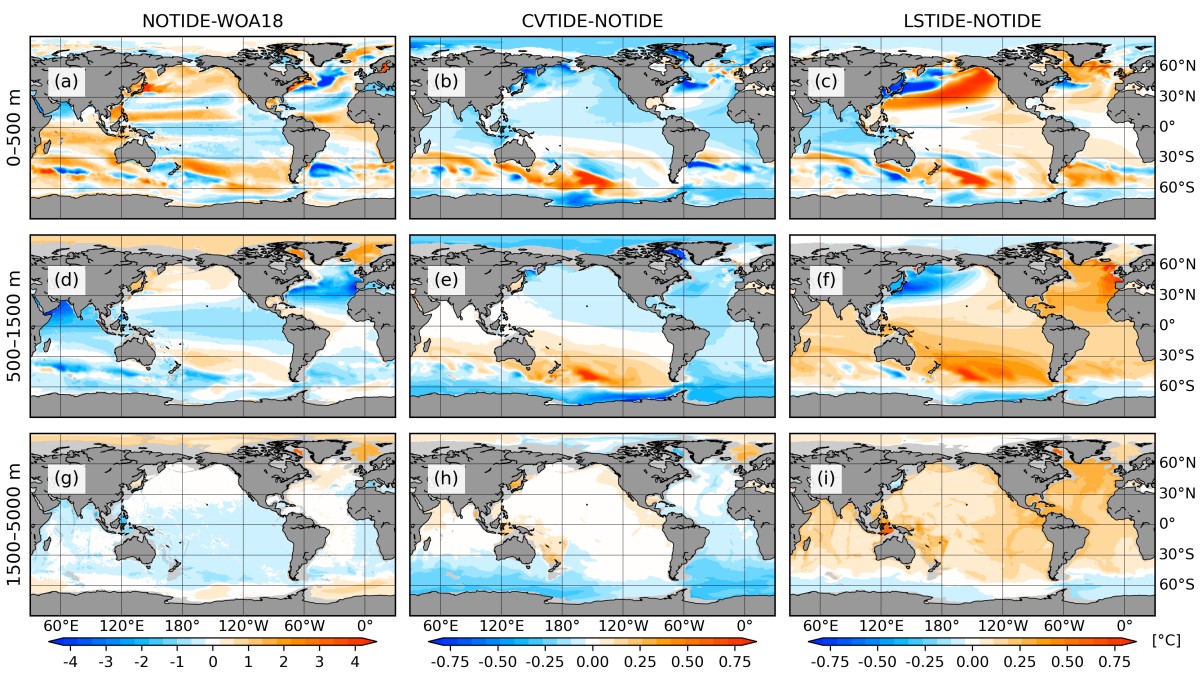

**Figure 3.** Depth averaged temperature biases between the model results and WOA18 in the upper (0–500 m), intermediate (500–1500 m), and deep (1500–5000 m) ocean. The left column shows the biases between NOTIDE and WOA18, while the middle/right column shows the difference between CVTIDE/LSTIDE and NOTIDE. Opposite colours in the left column and the middle/right column indicate CVTIDE/LSTIDE reduces temperature biases in NOTIDE.

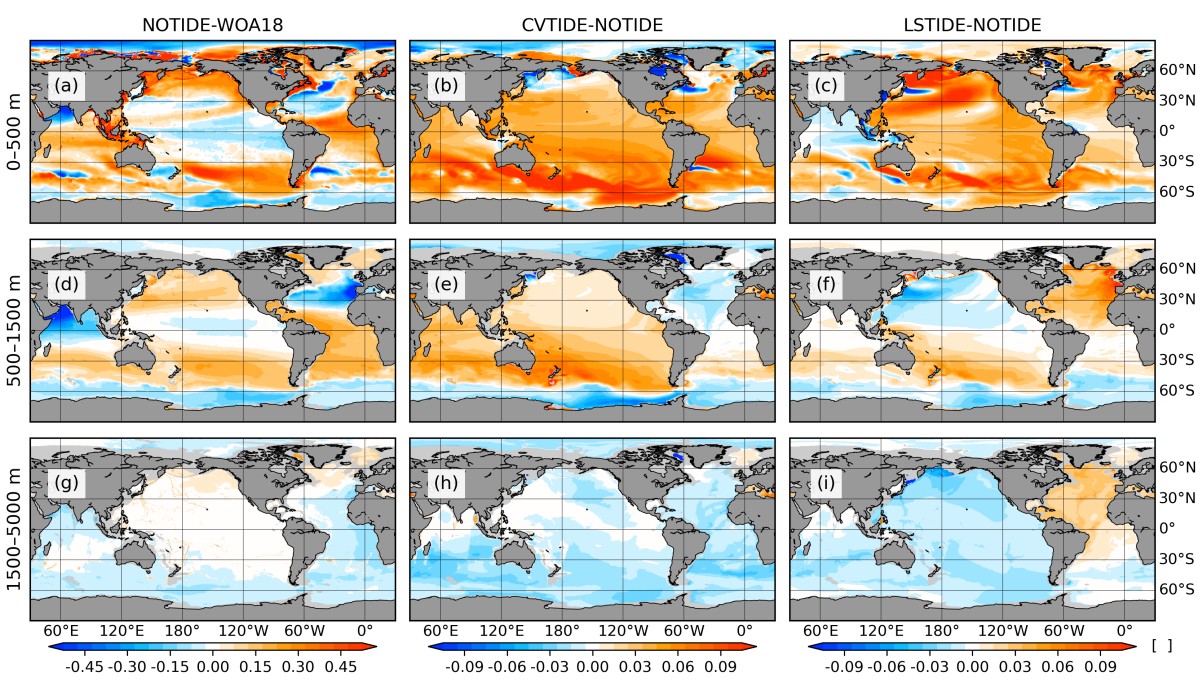

**Figure 4.** The same as Fig. 3, but for salinity.





**Figure 5.** Basin averaged hydrographic biases between the model results and WOA18 with respect to water depths. The hydrographic biases are absolute values. The left and the right column represent temperature and salinity biases, while each row shows the biases in different basins. Note that the averaging is nodal area-weighted.

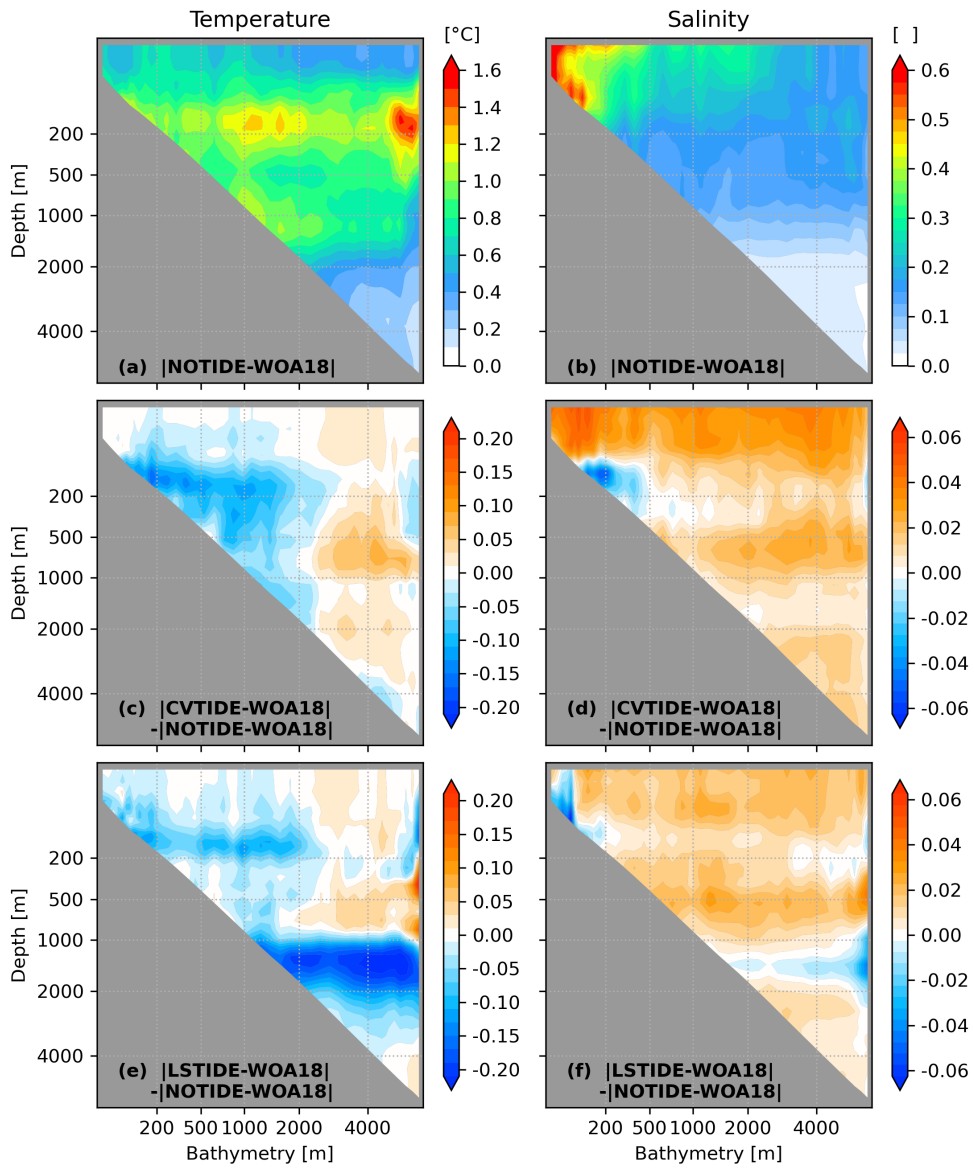

**Figure 6.** Bathymetry averaged hydrographic biases between the model results and WOA18 with respect to water depths. The hydrographic biases are absolute values, and model mesh grids with the same bathymetry (maximum water depth) are assembled and averaged. The middle and lower rows are presented as the difference with the upper row; thus the blue/red colours indicate tidal effects are decreasing/increasing model biases in the control run. Note that the averaging is nodal area-weighted, and the axes are scaled with cube root.

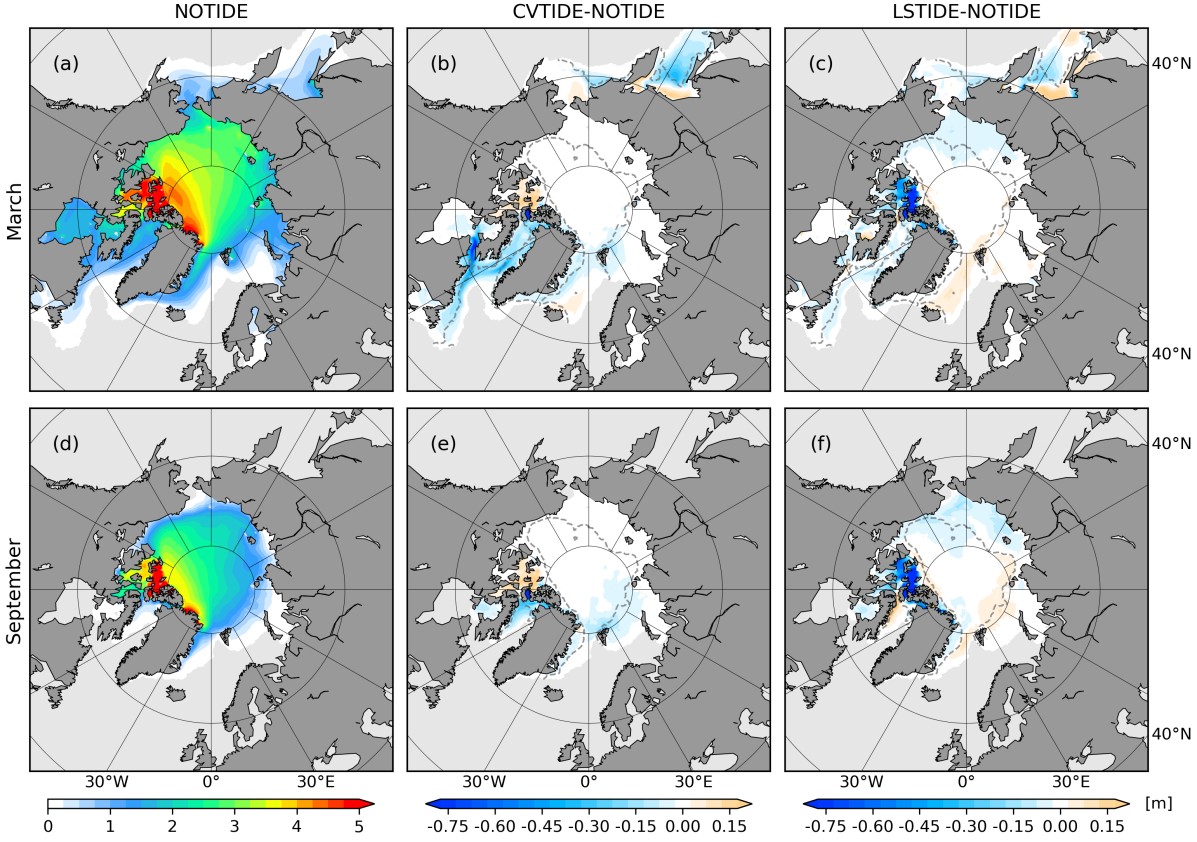

**Figure 7.** Arctic sea ice thickness in the three sensitivity runs. The left column shows the result from the control run, while the middle/right column shows the biases between CVTIDE/LSTIDE and NOTIDE. The upper and lower rows represent the model results in March and September, respectively. Note that the grey backgrounds mask the ice-free areas, and the grey dashed line represents the 500 m isobath.



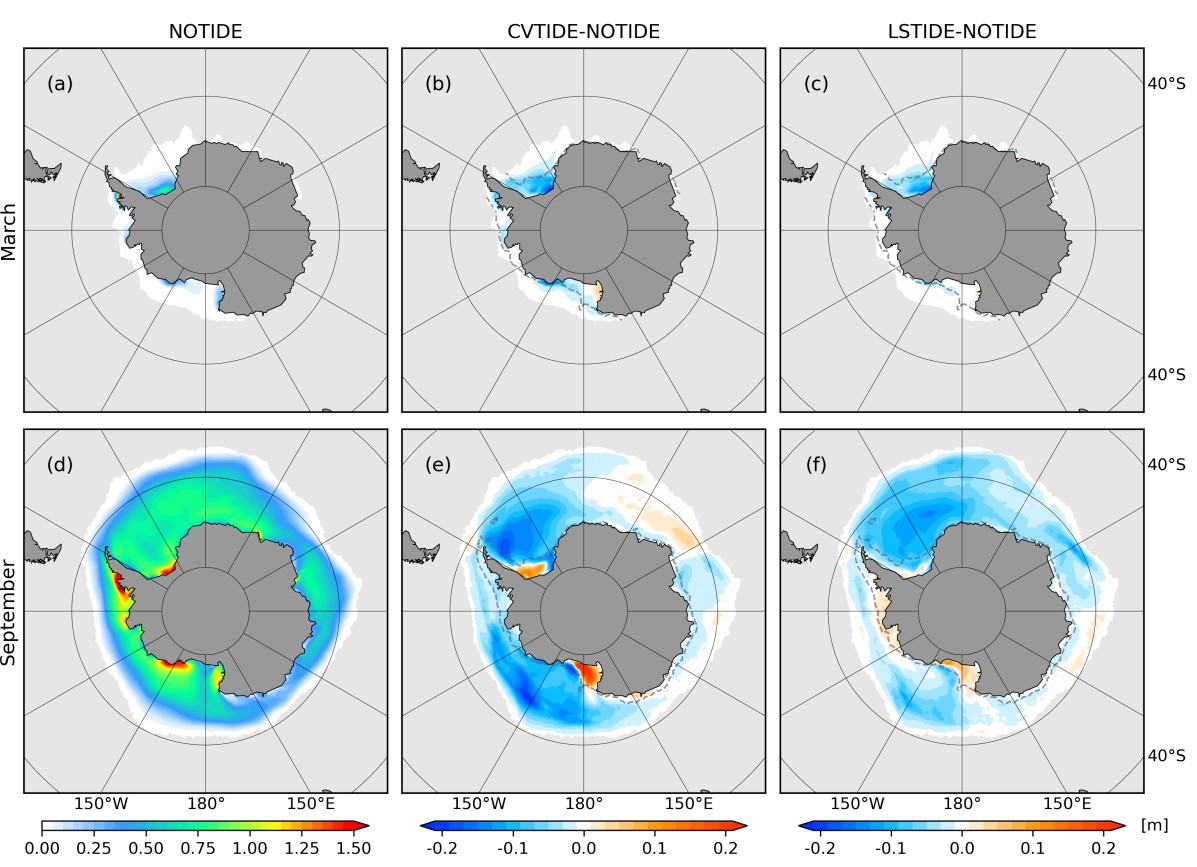

**Figure 8.** The same as Fig. 7, but for Antarctica. The grey dashed line represents the 1000 m isobath.



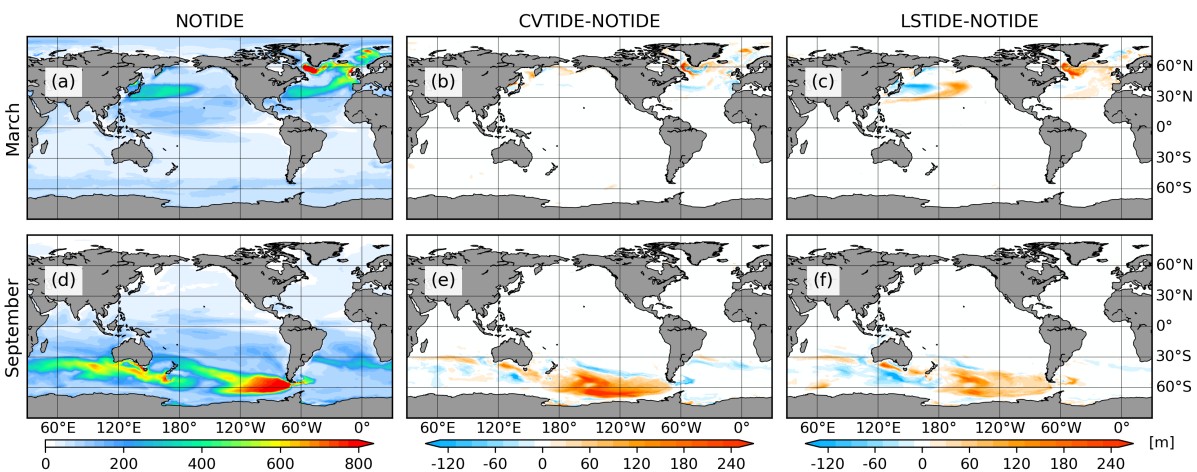

**Figure 9.** Mixed layer depths defined in Large et al. (1997). The left column represents the results from the control run, while the middle/right column represents the difference between CVTIDE/LSTIDE and NOTIDE. The upper and lower rows represent the model results in March and September, respectively.





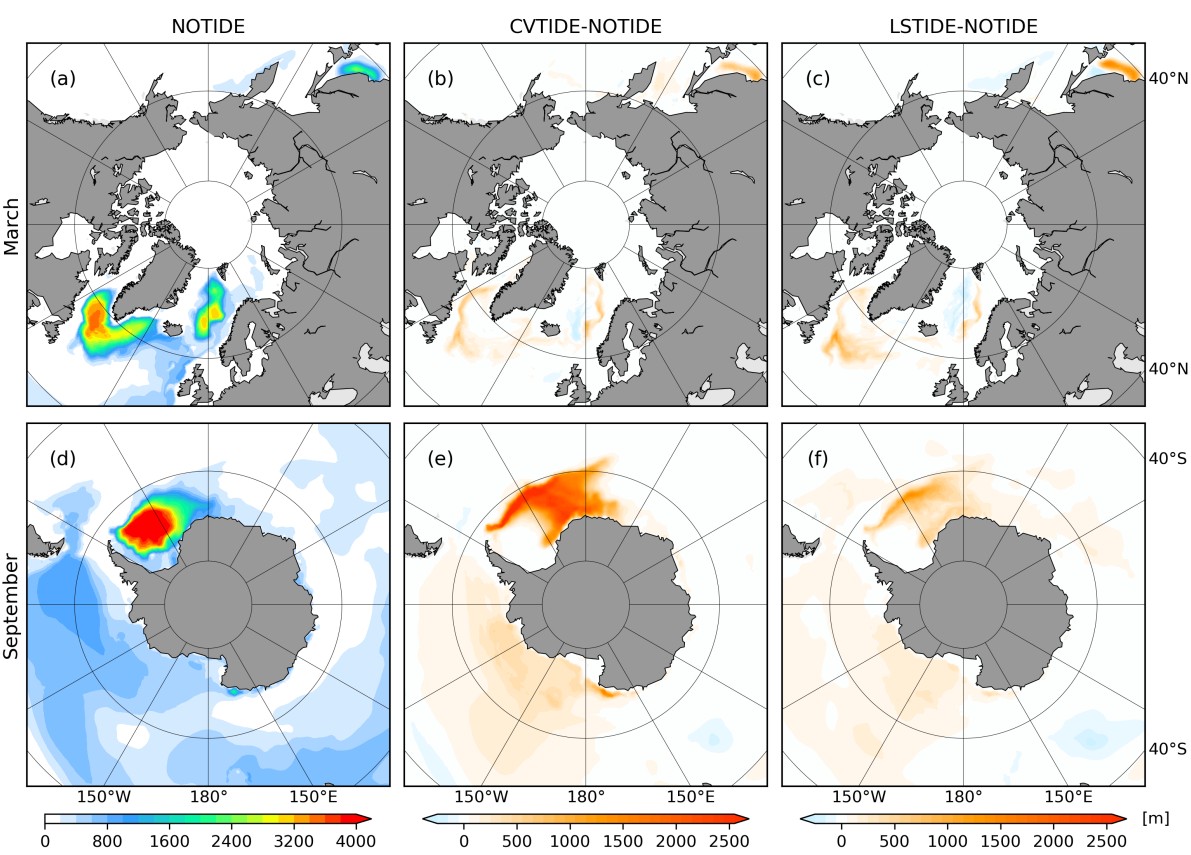

**Figure 10.** The same as Fig. 9 but for polar projection. Here the mixed layer depths are defined after Monterey and Levitus (1997).

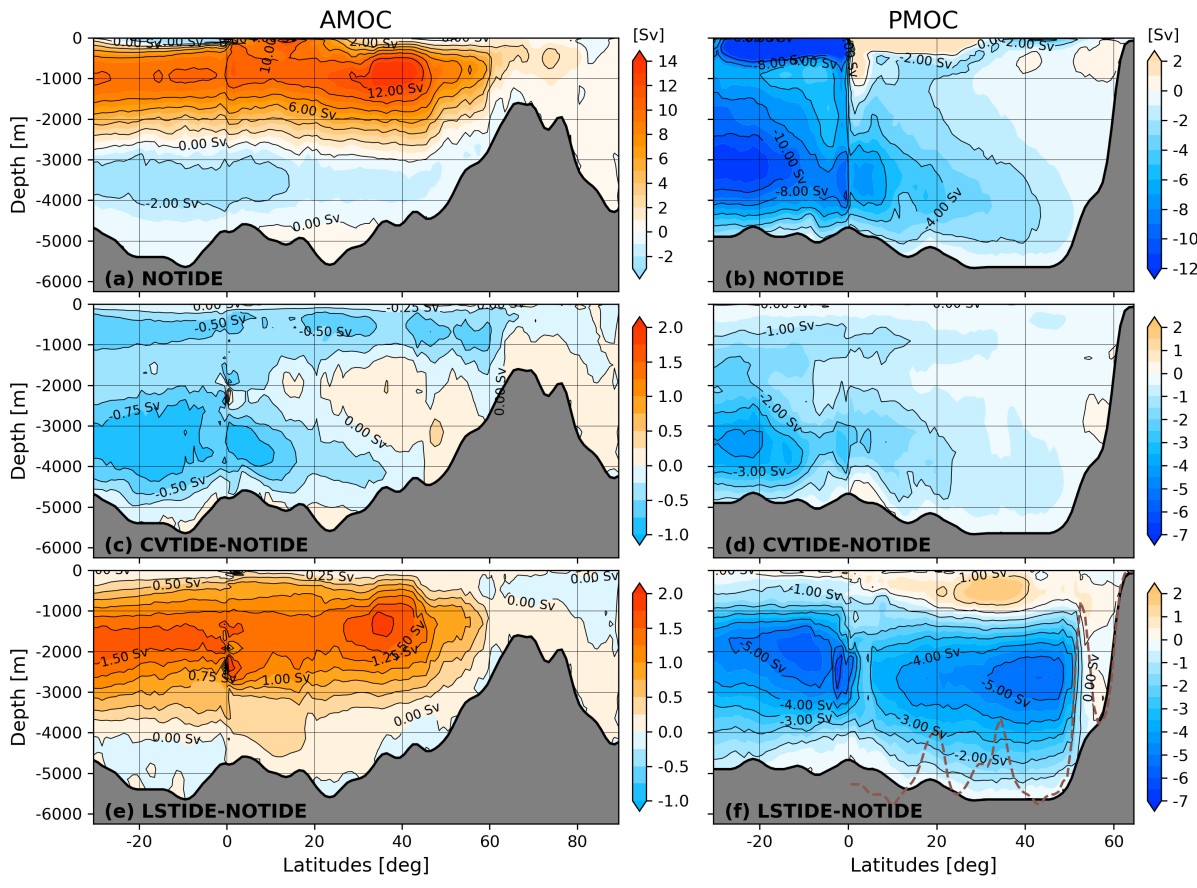

**Figure 11.** The MOC results in the three sensitivity runs. The upper panel shows the result from the control run, while the other two panels show the difference between CVTIDE/LSTIDE and NOTIDE. The left column and right column represent AMOC and PMOC, respectively. The brown dashed line in panel (f) denotes the North Pacific topography along the $180°$ transection. Note that the MOC contains the vertical component of bolus velocities (Gent and Mcwilliams, 1990). The calculation of MOC in the unstructured mesh applies the algorithm introduced in Sidorenko et al. (2020).



**Figure 12.** Depth averaged vertical diffusivity in the upper (0–500 m), intermediate (500–1500 m), deep (1500–5000 m) ocean, and the bottom layer of the ocean. The left column shows vertical diffusivity in the control run, while the middle/right column shows the difference between CVTIDE/LSTIDE and NOTIDE. Note that vertical diffusivity is shown in a decimal logarithm scale, and the middle/right column shows the difference of scaled vertical diffusivity.

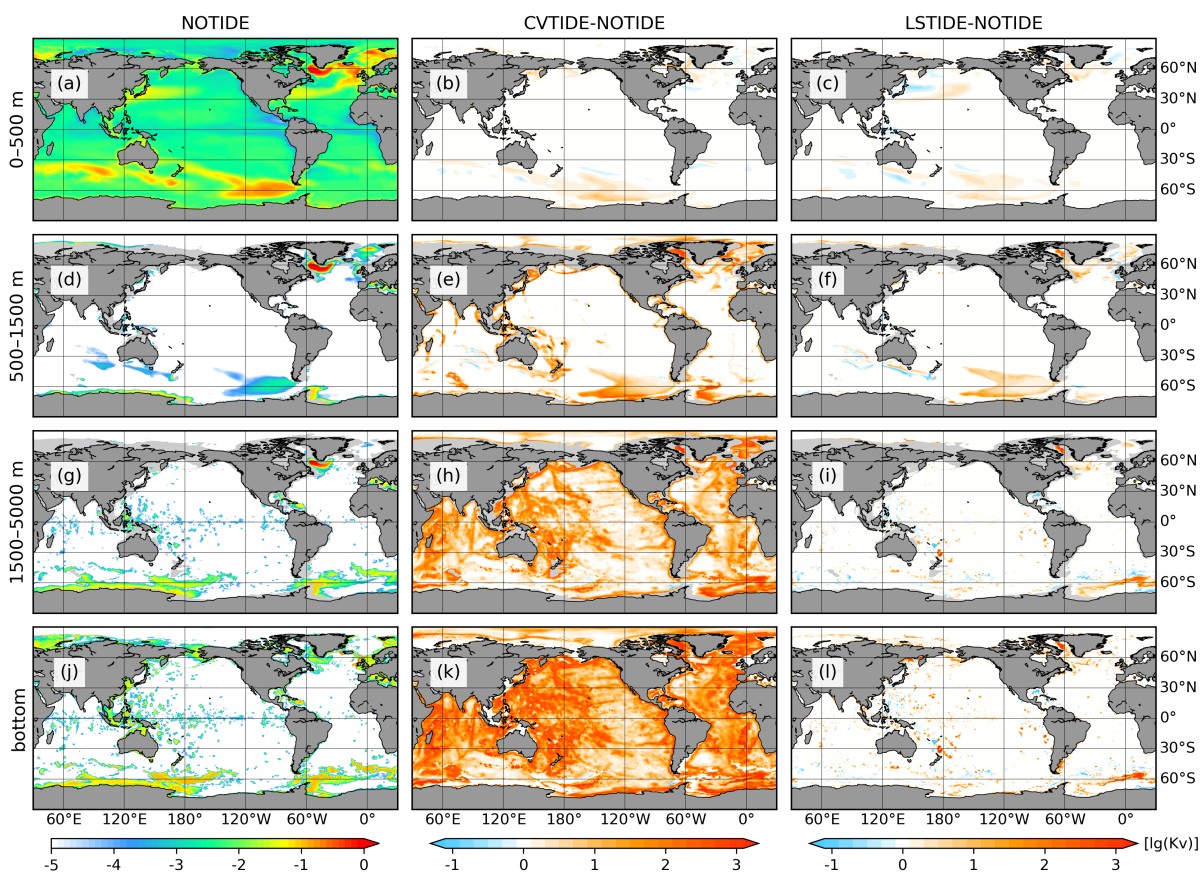



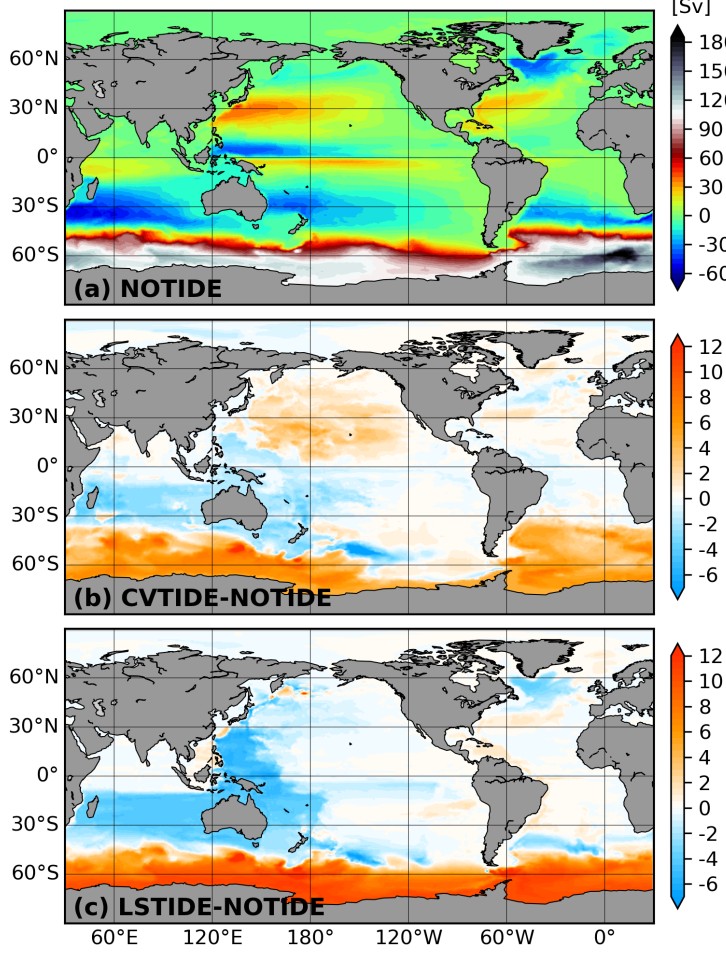

**Figure 13.** The barotropic streamfunction in the three sensitivity runs. The upper panel shows the result in the control run, while the middle/lower panel shows the difference between CVTIDE/LSTIDE and NOTIDE. Note that the lower panel also represents the tidal residual mean circulation. The calculation of barotropic streamfunction in the unstructured mesh applies the algorithm introduced in Sidorenko et al. (2020).



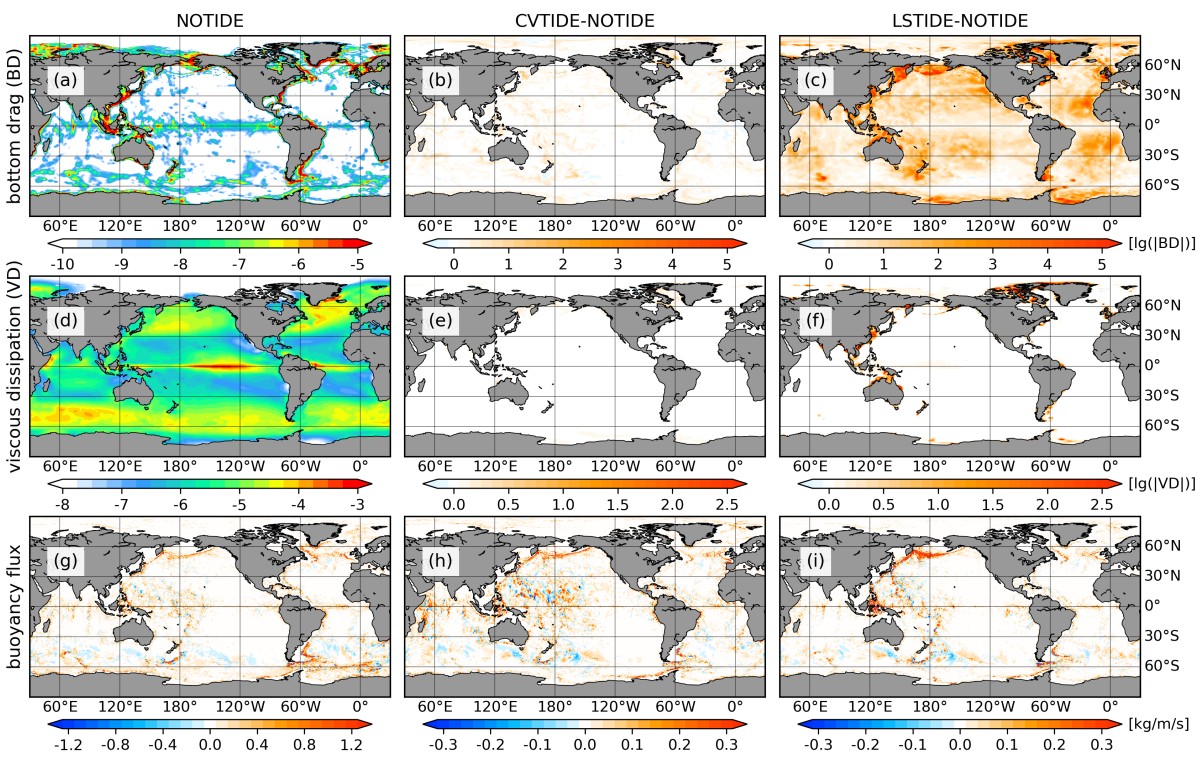

**Figure 14.** Horizontal distribution of vertically integrated energy diagnostic terms in Table 3. The upper, middle, and lower rows indicate viscous dissipation, bottom drag, and buoyancy flux, respectively. The left column shows the control run, and the middle/right column shows the difference between CVTIDE/LSTIDE and NOTIDE. Note that the upper and middle rows are shown in a decimal logarithm scale, which is the same as Fig. 12.





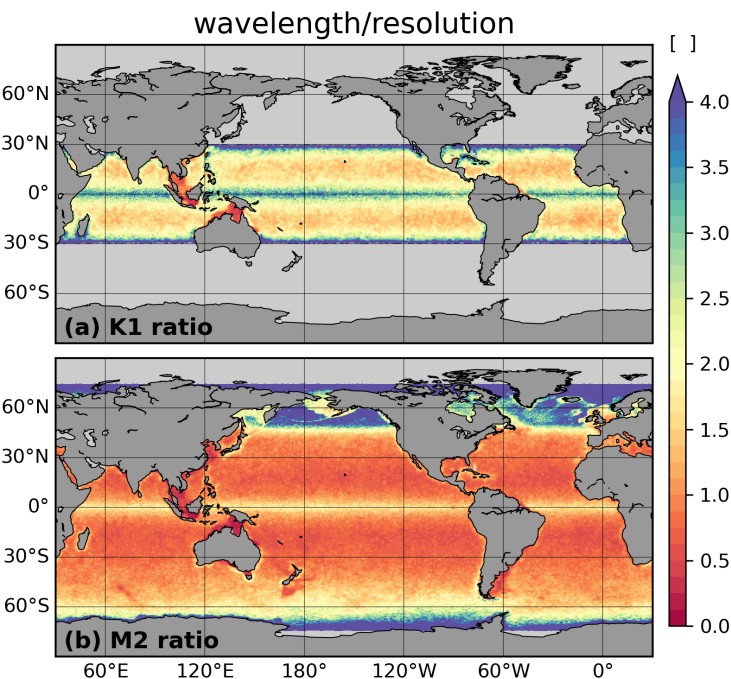

**Figure 15.** The ratio of mode-1 internal tide wavelength and mesh resolution. The upper and lower panels represent the ratio for K1 and M2 internal tides, respectively. Note here that the wavelengths are calculated by solving a Sturm–Liouville eigenvalue problem, which can be referred to in Song and Chen (2020, their Eq. 16).

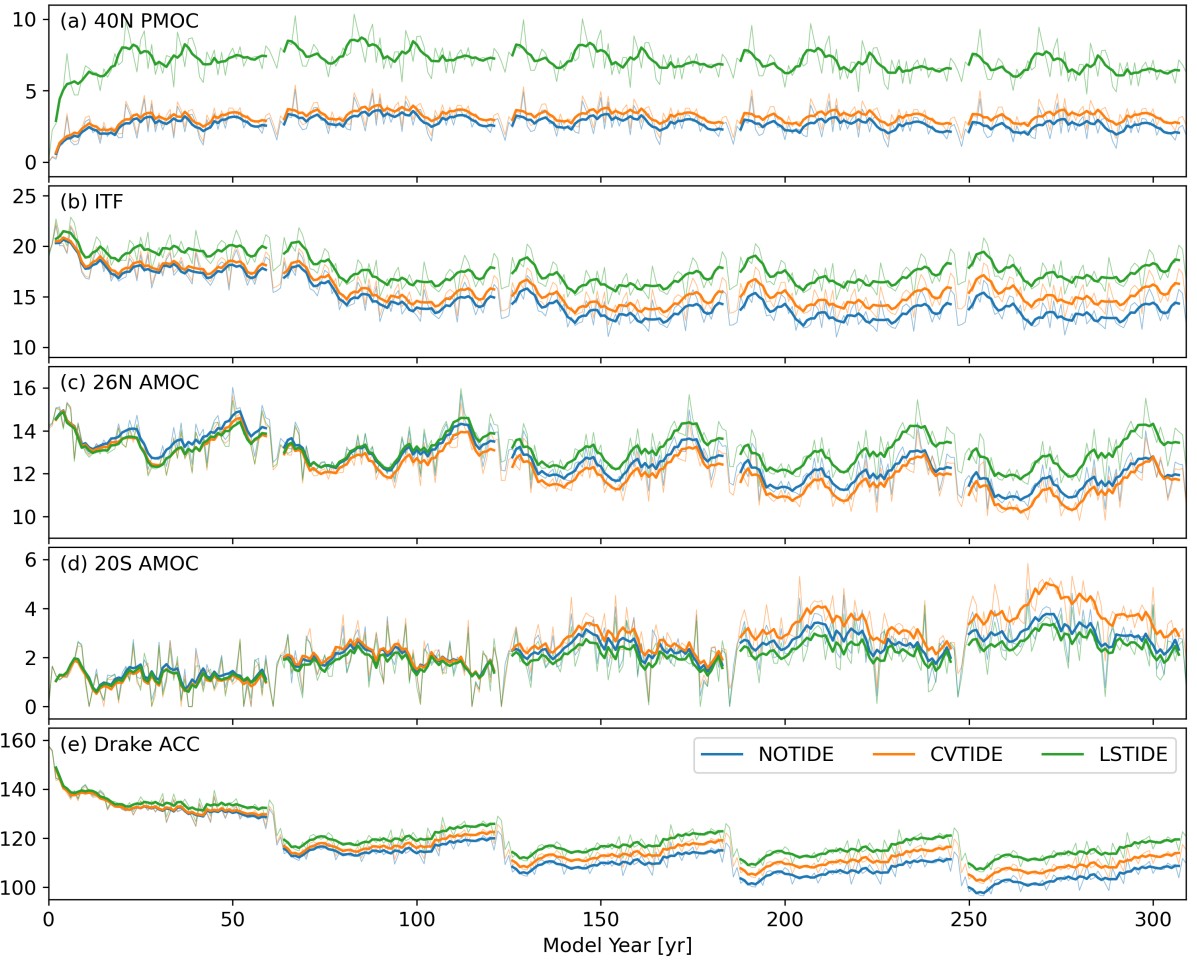

**Figure 16.** The time series of (a) 40° N PMOC, (b) ITF, (c) 26° N AMOC, (d) 20° S AMOC, (e) ACC through the Drake Passage from the three sensitivity runs in all five cycles. The light and thin lines are original annual time series, and the dark and thick lines are five-year moving average results for each cycle. All vertical axes have the unit of Sv. Note that 26° N AMOC and 20° S AMOC denote the strength of AMOC upper cell and lower cell, respectively. The calculation of ITF and ACC flux through the transections in the Indonesian Archipelago and the Drake Passage applies the algorithm introduced in Sidorenko et al. (2020).



**Figure 17.** The GMOC and potential density with reference depth at 4000 m ($\sigma_4$) in the southern hemisphere. The upper panels denote results in the control run, while the middle/lower panels denote the difference between CVTIDE/LSTIDE and NOTIDE. Note that the potential density in the figure is globally zonal averaged, with meridional bin set to $1°$.





**Table 1.** Basic information of the sensitivity runs.

| Experiment ID | Vertical mixing scheme | Tidal motions | Description |
|:---:|:---:|:---:|:---:|
| NOTIDE | KPP | False | FESOM2 control run |
| CVTIDE | KPP+CVMIX_TIDAL | False | FESOM2 considering tide-induced mixing param. |
| LSTIDE | KPP | True | FESOM2 considering lunisolar tidal motions |



**Table 2.** Model spin-up strategy for each sensitivity run.

| Cycle ID | R1 | R2 | R3 | R4 | R5 |
|---|---|---|---|---|---|
| Forcing year | 1948–2009 | 1948–2009 | 1948–2009 | 1948–2009 | 1948–2009 |
| Model year | 1–62 | 63–124 | 125–186 | 187–248 | 249–310 |





**Table 3.** Global integration of several energy diagnostic terms from the model results. The surface energy input, bottom drag, viscous dissipation, buoyancy flux, barotropic tide power, and kinetic energy terms are expressed as $\tau_{\mathbf{s}} \cdot \mathbf{u_s}$, $\tau_{\mathbf{b}} \cdot \mathbf{u_b}$, $-\rho_0 A_v \left( \partial \mathbf{u_h} / \partial z \right)^2$, $\rho w$, $\rho_0 \mathbf{u_h} \cdot \nabla \Omega$ and $\rho_0 \mathbf{u_h}^2 / 2$, respectively. $\mathbf{u_h}$ and $w$ represent the horizontal and vertical components of velocities; $\mathbf{u_s}$ and $\mathbf{u_b}$ are $\mathbf{u_h}$ at ocean surface and bottom; $\tau_{\mathbf{s}}$ and $\tau_{\mathbf{b}}$ are surface stress and bottom stress; $A_v$ is vertical viscosity; $\rho$ and $\rho_0$ are seawater density and reference seawater density; $\Omega$ is the tidal potential term. Surface energy input, bottom drag, viscous dissipation, and barotropic tide power have the unit of $\mathrm{kg\ m^2\ s^{-3}}$, buoyancy flux has the unit of $\mathrm{kg\ m\ s^{-1}}$ and kinetic energy has the unit of $\mathrm{kg\ m^2\ s^{-2}}$. All terms are calculated at each timestep and then averaged over time (take buoyancy flux as an example, $\overline{\rho w}$ instead of $\overline{\rho}\,\overline{w}$).

| Experiment ID | surface energy input | bottom drag | viscous dissipation | buoyancy flux | barotropic tide power | kinetic energy |
|:---:|:---:|:---:|:---:|:---:|:---:|:---:|
| NOTIDE | $3.11 \times 10^{12}$ | $-1.80 \times 10^{11}$ | $-2.19 \times 10^{12}$ | $1.68 \times 10^{11}$ | $0.00 \times 10^{12}$ | $1.30 \times 10^{18}$ |
| CVTIDE | $3.11 \times 10^{12}$ | $-1.84 \times 10^{11}$ | $-2.19 \times 10^{12}$ | $1.82 \times 10^{11}$ | $0.00 \times 10^{12}$ | $1.31 \times 10^{18}$ |
| LSTIDE | $3.11 \times 10^{12}$ | $-8.54 \times 10^{11}$ | $-2.29 \times 10^{12}$ | $1.85 \times 10^{11}$ | $4.69 \times 10^{12}$ | $1.53 \times 10^{18}$ |