# Peer review of "The tidal effects in the Finite-volumE Sea ice—Ocean Model (FESOM2.1): a comparison between parameterised tidal mixing and explicit tidal forcing"

_Geoscientific Model Development, 2022_

## Referee Comment (RC1)

**Review of "The tidal effects in the FInite-volumeE Sea ice-Ocean Model (FESOM2.1): a comparison between parameterised tidal mixing and explicit tidal forcing" by Song et al**

Jonas Nycander

This manuscript compares three different simulations with an ocean general circulation model: a control simulation without the effect of tides (NOTIDE), a simulation with parameterised tidal mixing (CVTIDE), and a simulation with explicit tidal forcing (LSTIDE). The differences are in general rather small. Perhaps the most interesting and significant ones are seen in the meridional overturning in Fig 11. CVTIDE has a stronger deep cell involving AABW, while LSTIDE has a stronger AMOC and a stronger cell in the North Pacific, with upwelling around 50°N.

The are major problems, both with the setup of the simulations and with the analysis of the results, and I therefore do not recommend publication of this manuscript.

Detailed comments:

- The tidal parameterisation in CVTIDE is very crude. It is based on a simple scaling estimate of the tidal generation of internal waves, but, for example, does not distinguish regions where these waves are trapped or propagating. There are more serious calculations of the tidal generation of internal waves by, for example, de Lavergne et al. (2019), with data available at https://www.seanoe.org/data/00470/58153/.

- As noted in the manuscript, the horizontal resolution is insufficient to resolve the internal tides in most of the ocean. It is therefore clear already from the outset that the tidal mixing can not be captured correctly. It should also be remembered that resolving the generation of internal tides is a necessary but not sufficient condition to describe tidal mixing. To do that, you must also describe the breaking of the internal waves, not just their generation. If, because of insufficient

resolution, they decay by viscous dissipation instead of by breaking, the mixing is not captured.

- It is unclear how the tidal motion in LSTIDE leads to vertical mixing, i.e. larger vertical diffusivity.

- In Figs 3-6 the hydrography in the simulations is compared to the observationally based data set WOA18. The results are mixed, and the differences between the different simulations are generally small compared to the bias of the control simulation. It is clear that neither CVTIDE nor LSTIDE gives any decisive improvement, and the improvements that exist in som regions may well be for the wrong reason. For example, biases caused by the background diffusivity, the K-profile parameterisation or the GM-parameterisation might be compensated by the parameterisation of tidal mixing. It is therefore difficult to draw any conclusion at all from these figures. CVTIDE and LSTIDE should instead be regarded as sensitivity tests, and the interesting question is not whether they decrease the bias, but how they modify the hydrography compared to NOTIDE.

- The key to understanding the effect on hydrography is the vertical diffusivity. Its geographical distribution is shown in Fig 12, but this should be complemented by plots with the vertical profile of the diffusivity, along with the vertical hydrographic profiles.

- In my view, the clearest and most interesting effect in LSTIDE is the strongly increased strength of the overturning cell in the North Pacific seen in Fig 11f. What evidence shows that this is an improvement?

- To explain the increased strength of the overturning cell in the North Pacific, the authors invoke the increased vertical diffusivity at the Kuril Ridge and Aleutian Ridge seen in Fig 12l and probably caused by trapped internal tides. This might be correct, but no strong support for this explanation is shown. Here are some problems. i) According to Fig 12 the vertical diffusivity in the Northern Pacific increases much more in CVTIDE than in LSTIDE, and yet there is much less upwelling in CVTIDE. In order to make the explanation credible, the pattern of vertical diffusivity (geographical and vertical) in the Northern Pacific should be studied in detail. ii) The strong diffusivity could be caused by resonant trapped waves, but it could also be caused simply by strong shear of the barotropic tide caused by bottom friction on

the continental shelf. It should be possible to check which alternative is correct.

- In section 5.2 it is argued that the stronger AMOC in LSTIDE is caused by increased upwelling in North Pacific and the Indonesian Archipelago. This seems far-fetched. An alternative explanation is that it is caused by increased vertical diffusivity in upper 3000 m of the Atlantic itself, but this is difficult to judge since vertical diffusivity profiles are not shown.

- The energy diagnostics in Table 3 are potentially interesting, but unfortunately incomplete. The surface energy input, bottom drag, viscous dissipation, buoyancy flux and barotropic tide power are terms in the budget for kinetic energy. However, the budget is far from closed, particularly in LSTIDE. (Note that the sign of the buoyancy flux should be changed when calculating the budget, since a positive $\overline{\rho w}$ is a conversion from kinetic energy to potential energy, i.e. a sink of kinetic energy.) It is striking that the barotropic tide power in LSTIDE is much larger than the increase of the sinks. The main missing term is probably energy loss due to horizontal viscosity, which should therefore also be diagnosed. If there are no more missing terms, the remaining residual will then be due to numerical errors. This is essential to know.

**References**

de Lavergne, C., S. Falahat, G. Madec, F. Roquet, J. Nycander and C. Vic, 2019: Toward global maps of internal tide energy sinks. *Ocean Model.*, **137**, 52–75.

---

## Author Comment (AC2)

**Authors' Response to Reviews of**

**The tidal effects in the Finite-volumE Sea ice–Ocean Model (FESOM2.1): a comparison between parameterised tidal mixing and explicit tidal forcing**

Pengyang Song et al.

*Geosci. Model Dev. Discuss.,* `10.5194/gmd-2022-25`
* * *
**RC:** *Reviewers' Comment*,    AR: Authors' Response,    ☐ Manuscript Text

**1. Reviewer #1**

**1.1. General comments**

**RC:** *This manuscript compares three different simulations with an ocean general circulation model: a control simulation without the effect of tides (NOTIDE), a simulation with parameterised tidal mixing (CVTIDE), and a simulation with explicit tidal forcing (LSTIDE). The differences are in general rather small. Perhaps the most interesting and significant ones are seen in the meridional overturning in Fig 11. CVTIDE has a stronger deep cell involving AABW, while LSTIDE has a stronger AMOC and a stronger cell in the North Pacific, with upwelling around 50° N.*

*There are major problems, both with the setup of the simulations and with the analysis of the results, and I therefore do not recommend publication of this manuscript.*

**AR:** Thank you for reviewing this article and leaving the comments. Responses to your comments are listed below.

**1.2. Detailed comments**

**RC:** *The tidal parameterisation in CVTIDE is very crude. It is based on a simple scaling estimate of the tidal generation of internal waves, but, for example, does not distinguish regions where these waves are trapped or propagating. There are more serious calculations of the tidal generation of internal waves by, for example, de Lavergne et al. (2019), with data available at https://www.seanoe.org/data/00470/58153/.*

*de Lavergne, C., S. Falahat, G. Madec, F. Roquet, J. Nycander and C. Vic, 2019: Toward global maps of internal tide energy sinks. Ocean Model., 137, 52–75.*

**AR:** In our research, we have found modern databases which feature either higher resolution or better dynamic processes. However, the approaches these databases implemented into the model are the same. That is via changing the vertical diffusivity. In our work, we mainly focus on the comparison between changing model vertical diffusivity and adding real tidal motions. Thus, we use the "basic" Simmons' data for two reasons: (1) to make our CVTIDE model results comparable to other models which also implemented the CVMIX_TIDAL parameterisation. (2) to compare the two approaches of considering tidal effects in a model. As to the new databases, we consider applying them in the future to perfect the expression of tide-induced mixing in an ocean model.

**RC:** *As noted in the manuscript, the horizontal resolution is insufficient to resolve the internal tides in most of the ocean. It is therefore clear already from the outset that the tidal mixing can not be captured correctly. It should also be remembered that resolving the generation of internal tides is a necessary but not sufficient condition to describe tidal mixing. To do that, you must also describe the breaking of*

*the internal waves, not just their generation. If, because of insufficient resolution, they decay by viscous dissipation instead of by breaking, the mixing is not captured.*

AR: In our research, we cannot describe the breaking of internal waves in the model. We believe this is a common challenge to an Ocean General Circulation Model (OGCM). Klymak and Legg (2010) raised a parameterisation of internal wave breaking. However, this is not generally applied in an OGCM simulation. In our research, the mixing of tides is considered by the vertical shear of baroclinic tides.

Klymak, J. M., & Legg, S. M. (2010). A simple mixing scheme for models that resolve breaking internal waves. Ocean Modelling, 33(3-4), 224-234.

RC: *It is unclear how the tidal motion in LSTIDE leads to vertical mixing, i.e. larger vertical diffusivity.*

AR: The model applies the KPP scheme (Large et al., 1994), in which vertical shear is decisive to the mixing in the interior of the ocean. Thus, the vertical mixing in LSTIDE is considered by the vertical shear of baroclinic tides.

Large, W. G., McWilliams, J. C., & Doney, S. C. (1994). Oceanic vertical mixing: A review and a model with a nonlocal boundary layer parameterization. Reviews of geophysics, 32(4), 363-403.

RC: *In Figs 3-6 the hydrography in the simulations is compared to the observationally based data set WOA18. The results are mixed, and the differences between the different simulations are generally small compared to the bias of the control simulation. It is clear that neither CVTIDE nor LSTIDE gives any decisive improvement, and the improvements that exist in some regions may well be for the wrong reason. For example, biases caused by the background diffusivity, the K-profile parameterisation or the GM-parameterisation might be compensated by the parameterisation of tidal mixing. It is therefore difficult to draw any conclusion at all from these figures. CVTIDE and LSTIDE should instead be regarded as sensitivity tests, and the interesting question is not whether they decrease the bias, but how they modify the hydrography compared to NOTIDE.*

AR: Thanks for this comment. We agree that KPP and GM might be compensated by either CVMIX_TIDAL scheme or tidal motions. We will change this expression and focus more on how they modify the model results instead of decreasing the biases.

RC: *The key to understanding the effect on hydrography is the vertical diffusivity. Its geographical distribution is shown in Fig 12, but this should be complemented by plots with the vertical profile of the diffusivity, along with the vertical hydrographic profiles.*

AR: Thanks for this comment. We will add the vertical profile figures in our revision.

RC: *In my view, the clearest and most interesting effect in LSTIDE is the strongly increased strength of the overturning cell in the North Pacific seen in Fig 11f. What evidence shows that this is an improvement?*

AR: Figure 3 shows that LSTIDE improves the temperature results because of the PMOC cell in the North Pacific. By comparing Fig. 3d and 3f, the intermediate layer of the North and Equatorial Pacific shows model biases in NOTIDE are reduced. By comparing Fig. 3g and 3i, the cold biases in the Pacific Ocean are reduced in LSTIDE. In Fig. 5c, the temperature biases in the Pacific Ocean is reduced at 1000–3500 m. Fig. 6e also shows significant temperature bias reducing below 1000 m.

RC: *To explain the increased strength of the overturning cell in the North Pacific, the authors invoke the increased vertical diffusivity at the Kuril Ridge and Aleutian Ridge seen in Fig 12l and probably caused by trapped internal tides. This might be correct, but no strong support for this explanation is shown. Here*

*are some problems. i) According to Fig 12 the vertical diffusivity in the Northern Pacific increases much more in CVTIDE than in LSTIDE, and yet there is much less upwelling in CVTIDE. In order to make the explanation credible, the pattern of vertical diffusivity (geographical and vertical) in the Northern Pacific should be studied in detail. ii) The strong diffusivity could be caused by resonant trapped waves, but it could also be caused simply by strong shear of the barotropic tide caused by bottom friction on the continental shelf. It should be possible to check which alternative is correct.*

AR: Thanks for this comment. As to point (1), we think that changing vertical diffusivity in the model has a much slower response compared to the tidal dynamics itself. Three hundred years might be long enough for dynamic processes, but not for thermodynamic processes. This is also why CVTIDE does not show mixing as strong as LSTIDE in the North Pacific. As to point (2), we cannot distinguish the two factors now and our research needs more add-on work.

RC: *In section 5.2 it is argued that the stronger AMOC in LSTIDE is caused by increased upwelling in North Pacific and the Indonesian Archipelago. This seems far-fetched. An alternative explanation is that it is caused by increased vertical diffusivity in upper 3000 m of the Atlantic itself, but this is difficult to judge since vertical diffusivity profiles are not shown.*

AR: Thanks for this comment. We will add the diagnostic of vertical profile in the Atlantic Ocean in our revision.

RC: *The energy diagnostics in Table 3 are potentially interesting, but unfortunately incomplete. The surface energy input, bottom drag, viscous dissipation, buoyancy flux and barotropic tide power are terms in the budget for kinetic energy. However, the budget is far from closed, particularly in LSTIDE. (Note that the sign of the buoyancy flux should be changed when calculating the budget, since a positive $\rho w$ is a conversion from kinetic energy to potential energy, i.e. a sink of kinetic energy.) It is striking that the barotropic tide power in LSTIDE is much larger than the increase of the sinks. The main missing term is probably energy loss due to horizontal viscosity, which should therefore also be diagnosed. If there are no more missing terms, the remaining residual will then be due to numerical errors. This is essential to know.*

AR: Thanks for this comment. We will complete the energy diagnostic parts in our revision, especially horizontal viscosity and kinetic–potential energy conversion.

---

## Author Comment (AC3)

**Authors' Response to Reviews of**

**The tidal effects in the Finite-volumE Sea ice–Ocean Model (FESOM2.1): a comparison between parameterised tidal mixing and explicit tidal forcing**

Pengyang Song et al.
*Geosci. Model Dev. Discuss.,* `10.5194/gmd-2022-25`
* * *
**RC:** *Reviewers' Comment*,     AR: Authors' Response,     ☐ Manuscript Text

**1. Reviewer #2**

**1.1. General comments**

**RC:** *This manuscript deals with a comparison of a coarse resolution ocean model with a very simplistic parameterisation of tidal mixing and a model version without that scheme but explicit tidal potential included.*

*There is a major physical flaw in this model setup. The authors seem to believe that having the long internal tidal waves in their model they have implemented "tidal mixing". This is not the case. The long tidal waves do not break, except maybe close to continental shelves. In contrast, it is the energy transfer by non-linear interaction from the long internal tides to the shorter background internal wave field, the "GM wave field", and the breaking of the shortest wave lengths in that GM wave field, which leads to the bulk of the "tidal mixing" in the interior ocean. This process is certainly not resolved in the current model. One would need very high resolution and non-hydrostatic dynamics to simulate that process. In the present manuscript, it is unclear how the long tidal waves would induce mixing, and the issue is not touched at all. However, it is clear that it would not correspond in any way to what happens in the ocean.*

*It is therefore of no use to compare the model experiments as done here, and this manuscript should be rejected.*

AR: Thanks for reviewing this paper. We admit that our work does not fully consider the energy chain of tides. However, this is always a big challenge to simulate all-scale processes at a global range. The KPP scheme (Large et al., 1994) is applied in this model, where the mixing is shear-dependent. Thus, the baroclinic tides can lead to mixing in the model. In addition, our results show strong mixing caused by the trapped internal tides in the Kuril–Aleutian Ridge. That strong mixing causes stronger PMOC cell and shows better model performance in the Pacific Ocean. This is an important contribution of our work.

Large, W. G., McWilliams, J. C., & Doney, S. C. (1994). Oceanic vertical mixing: A review and a model with a nonlocal boundary layer parameterization. Reviews of geophysics, 32(4), 363-403.

---

## Author Response (AR1)

**Authors' Response to Reviews of**

**The tidal effects in the Finite-volumE Sea ice–Ocean Model (FESOM2.1): a comparison between parameterised tidal mixing and explicit tidal forcing**

Pengyang Song et al.

*Geosci. Model Dev. Discuss.,* `10.5194/gmd-2022-25`
* * *
**RC:** *Reviewers' Comment*,     AR: Authors' Response,     ☐ Manuscript Text

Dear editor and reviewers,

Thanks for the comments on our manuscript. By improving our work and revising our paper following the reviewers' guidance, we found that our last responses to some reviewers' comments are not sufficient. Thus, we want to add some responses to the reviewers' comments to better explain and improve our work.

Yours sincerely,

Pengyang Song et al.

**1.   Reviewers' comments**

**RC:** *It is unclear how the tidal motion in LSTIDE leads to vertical mixing, i.e. larger vertical diffusivity.*

AR:   The model applies the KPP scheme, in which vertical shear is decisive to the mixing in the interior of the ocean. Thus, we thought the vertical mixing in LSTIDE is contained with the vertical shear of baroclinic tides.

However, by plotting the meridional transection of the vertical diffusivity from the model results, we found that vertical diffusivity is not significantly changed. The reason is that the model resolution can hardly simulate propagating internal tides, so is the vertical shear from internal tides. By studying how the upwelling occurs at the Kuril–Aleutian Ridge, we realised that the direct tide–topography interaction affects the along-boundary upwelling via the turbulent buoyancy flux. Referring to the Reynolds stress equation, the vertical diffusivity coefficient represents the strength of the turbulent buoyancy flux, which cannot be resolved by the model (Appendix A2 in the revised manuscript). But LSTIDE simulates tidal motions in the model and shows intense turbulent buoyancy flux at the Kuril–Aleutian Ridge, equal to enhancing vertical diffusivity in the model (Fig. 4). The equivalent vertical diffusivity caused by turbulent buoyancy fluxes can reach $10^{-2}\ m^2\ s^{-1}$. Unlike CVTIDE that assumes tidal mixing decaying from bottom to top, in LSTIDE the upwelling caused by turbulent buoyancy flux only occurs at near-bottom layers. That explains why the PMOC pattern in the North Pacific (Fig. 5f) coincides well with the bathymetry. The pattern of tide-induced upwelling which occurs near bottom also agrees with the argument raised in Ferrari et al. (2016).

Ferrari, R., Mashayek, A., McDougall, T. J., Nikurashin, M., & Campin, J. M. (2016). Turning ocean mixing upside down. Journal of Physical Oceanography, 46(7), 2239-2261.

**RC:** *In Figs 3-6 the hydrography in the simulations is compared to the observationally based data set WOA18. The results are mixed, and the differences between the different simulations are generally small compared to the bias of the control simulation. It is clear that neither CVTIDE nor LSTIDE gives any decisive improvement, and the improvements that exist in some regions may well be for the wrong rea-*

*son. For example, biases caused by the background diffusivity, the K-profile parameterisation or the GM-parameterisation might be compensated by the parameterisation of tidal mixing. It is therefore difficult to draw any conclusion at all from these figures. CVTIDE and LSTIDE should instead be regarded as sensitivity tests, and the interesting question is not whether they decrease the bias, but how they modify the hydrography compared to NOTIDE.*

AR: Below Fig. 1 and Fig. 2 show the meridional transection plot of hydrography and vertical diffusivity in the Atlantic Ocean and the Pacific Ocean. We can see from Figs. 1d, 1e, 2d and 2e that CVTIDE affects the ocean by enhancing the mixing in the interior ocean. Thus the patterns mainly feature two layers. Take ocean temperature as an example, a direct explanation is that with mixing enhanced in the ocean, the upper ocean would be cooler and the deeper ocean would be warmer. But this is different for LSTIDE, because LSTIDE does not introduce enough mixing. The hydrography in LSTIDE is also changed significantly due to the strengthening of PMOC–AMOC cycle. This is also the main finding of our work. In Fig. 2g, the reduction of temperature biases in NOTIDE is evident. And from our result, we think that with the vertical mixing scheme treated properly, the hydrography biases would be reduced via both interior mixing (like CVTIDE) and the adjustment of the MOC (like LSTIDE). That is why we talk about reducing the biases.

RC: *The key to understanding the effect on hydrography is the vertical diffusivity. Its geographical distribution is shown in Fig 12, but this should be complemented by plots with the vertical profile of the diffusivity, along with the vertical hydrographic profiles.*

AR: Below Fig. 1 and Fig. 2 show the transection plot of hydrography and vertical diffusivity in the Atlantic Ocean and the Pacific Ocean. These figures lead to the same conclusions as the figures we show in our manuscript. Therefore, we think it's better not to add too much redundant information to the manuscript.

RC: *To explain the increased strength of the overturning cell in the North Pacific, the authors invoke the increased vertical diffusivity at the Kuril Ridge and Aleutian Ridge seen in Fig 12l and probably caused by trapped internal tides. This might be correct, but no strong support for this explanation is shown. Here are some problems. i) According to Fig 12 the vertical diffusivity in the Northern Pacific increases much more in CVTIDE than in LSTIDE, and yet there is much less upwelling in CVTIDE. In order to make the explanation credible, the pattern of vertical diffusivity (geographical and vertical) in the Northern Pacific should be studied in detail. ii) The strong diffusivity could be caused by resonant trapped waves, but it could also be caused simply by strong shear of the barotropic tide caused by bottom friction on the continental shelf. It should be possible to check which alternative is correct.*

AR: Thanks again for this comment. In the model, vertical diffusivity coefficient ($K_v$) is dependent on the vertical shear. If the bottom friction provides strong vertical shear near the bottom, we should find evidence of larger vertical diffusivity along the ocean bottom. But we can see from the section plot of $K_v$ (Fig. 4c) that no significant change emerges in LSTIDE. This is because that the tidal velocity in deep ocean is not as large as in shallow ocean. Indeed, we agree that on continental shelves the bottom friction is strong and the strong vertical shear should increase bottom $K_v$ via enhanced vertical shear. However, this does not coincide with the upwelling in the North Pacific from 4000 m to 1000 m. Therefore, we rule out this possibility.

By checking the meridional transection plots of the vertical diffusivity and the turbulent buoyancy flux, we found that LSTIDE increases PMOC upwelling by introducing turbulent buoyancy flux instead of increasing $K_v$. The Appendix A2 of our revised manuscript explains that these two approaches are equivalent. If we also consider the equivalent vertical diffusivity caused by the turbulent buoyancy flux, we can find that the increase of vertical diffusivity in LSTIDE is much stronger than CVTIDE and is more concentrated at the near-bottom layers. That is why CVTIDE does not show the same changes as LSTIDE regarding the strength and shape of the North Pacific MOC cell.

**RC:** *In section 5.2 it is argued that the stronger AMOC in LSTIDE is caused by increased upwelling in North Pacific and the Indonesian Archipelago. This seems far-fetched. An alternative explanation is that it is caused by increased vertical diffusivity in upper 3000 m of the Atlantic itself, but this is difficult to judge since vertical diffusivity profiles are not shown.*

**AR:** We plot the Atlantic meridional section of vertical diffusivity in below Fig. 3. Our results do not show evidence of stronger vertical diffusivity in the mid- and low-latitude Atlantic Ocean in LSTIDE. Note that in high-latitude North Atlantic Ocean the large vertical diffusivity is applied for gravitational instability (deep convection). Thus, we don't think this should be the reason for a stronger AMOC in LSTIDE.

Another evidence to prove that the stronger AMOC in LSTIDE is not caused by the mixing in the Atlantic is that the pattern of AMOC difference does not show closed overturning circulation (streamfunction lines) in the mid-latitude Atlantic (Fig. 5e). If the the mixing in the Atlantic itself would explain the strengthening of the AMOC, the upwelling should happen in the Atlantic. However, we can only see a closed circulation pattern in the North Atlantic ($20°$ N–$40°$ N) with a strength of 0.5 $Sv$. This cannot explain the rest 1.5 $Sv$, which is the strength of the upper south-to-north flow and the lower north-to-south flow at the $30°$ S Atlantic.

**RC:** *The energy diagnostics in Table 3 are potentially interesting, but unfortunately incomplete. The surface energy input, bottom drag, viscous dissipation, buoyancy flux and barotropic tide power are terms in the budget for kinetic energy. However, the budget is far from closed, particularly in LSTIDE. (Note that the sign of the buoyancy flux should be changed when calculating the budget, since a positive $\rho w$ is a conversion from kinetic energy to potential energy, i.e. a sink of kinetic energy.) It is striking that the barotropic tide power in LSTIDE is much larger than the increase of the sinks. The main missing term is probably energy loss due to horizontal viscosity, which should therefore also be diagnosed. If there are no more missing terms, the remaining residual will then be due to numerical errors. This is essential to know.*

**AR:** Thanks again for pointing out this. We checked each term carefully and found that the buoyancy flux term is wrongly calculated. As to the LSTIDE, we missed the surface pressure power term, which counts for a large proportion of the tidal energy. As requested, we also calculated the horizontal viscous dissipation and added this to the table. We demonstrate the derivation and calculation of the kinetic energy budget in the Appendix A1 of the revised manuscript.

Below Table 1 shows that, from a global integration view, LSTIDE increases bottom drag and viscous dissipation terms by about 1 $TW$. The 4.5 $TW$ tidal potential power input is mostly balanced by the surface pressure power $-\mathbf{u_h} \cdot \nabla_h \left( \rho_0 g \eta \right)$. If we assume a high-resolution run, which involves propagating internal tides, this surface pressure power term would be smaller because internal tides decrease the surface tide amplitudes (Jayne and St. Laurent, 2001). Then this part of energy would be redistributed to buoyancy flux and viscous dissipation terms, depending on how much internal tide energy can be generated.

Jayne, S. R., & St. Laurent, L. C. (2001). Parameterizing tidal dissipation over rough topography. Geophysical Research Letters, 28(5), 811-814.

Table 1: A global integration of kinetic energy budget from the model results. The surface wind stress, bottom stress, vertical viscous dissipation, horizontal viscous dissipation, buoyancy flux, tidal potential power, surface pressure power terms and total kinetic energy are in order listed in the table. The expressions of these terms can be found in the Appendix A1 of the revised manuscript. Note that all except the last term have a unit of $TW$, while kinetic energy has a unit of $10^{18}$ $J$. This table is also added in the revised manuscript.

| Experiment ID | wind stress | bott stress | vert visc | horiz visc | buoy flux | tide | surf pressure | KE |
|---|---|---|---|---|---|---|---|---|
| NOTIDE | 3.11 | $-0.18$ | $-2.19$ | $-0.17$ | $-0.58$ | 0.00 | $-0.02$ | 1.30 |
| CVTIDE | 3.11 | $-0.18$ | $-2.19$ | $-0.18$ | $-0.58$ | 0.00 | $-0.02$ | 1.31 |
| LSTIDE | 3.11 | $-0.85$ | $-2.29$ | $-0.30$ | $-0.58$ | 4.55 | $-3.51$ | 1.53 |

[Figure]

Figure 1: Zonal mean meridional section of the Atlantic hydrography and vertical diffusivity. Note this is calculated with a zonal mean throughout the Atlantic basin. Panel (a) and (b) shows the model bias between NOTIDE and WOA18. The middle and bottom rows show the difference between CVTIDE/LSTIDE and NOTIDE. In the right column, vertical diffusivity is shown in a decimal logarithm scale.

[Figure]

Figure 2: The same as Fig. 1 but in the Indo-Pacific Ocean.

[Figure]

Figure 3: The vertical diffusivity, the turbulent buoyancy flux and its equivalent vertical diffusivity along the transection of 30° W. The calculation of turbulent buoyancy flux and the equivalent vertical diffusivity is introduced in the Appendix A2 of the revised manuscript. All panels are shown in a decimal logarithm scale.

[Figure]

Figure 4: The vertical diffusivity, the turbulent buoyancy flux and its equivalent vertical diffusivity along the transection of 165° E. The calculation of turbulent buoyancy flux and the equivalent vertical diffusivity is introduced in Appendix A2 of the revised manuscript. All panels are shown in a decimal logarithm scale. This figure is also added in the revised manuscript.

[Figure]

Figure 5: The MOC results in the three sensitivity runs. The upper panel shows the result from the control run, while the other two panels show the difference between CVTIDE/LSTIDE and NOTIDE. The left column and right column represent AMOC and PMOC, respectively. The brown dashed line in panel (f) denotes the North Pacific topography along the 180° transection. This figure is the same as Fig. 11 in the manuscript.

---

## Referee Report (RR1)

**Review of the first revision of "The tidal effects in the FInite-volumeE Sea ice-Ocean Model (FESOM2.1): a comparison between parameterised tidal mixing and explicit tidal forcing" by Song et al**

Jonas Nycander

There are some significant improvements in the revised manuscript, but the main problem remains. Over most of the global ocean, the resolution of the model is insufficient to resolve the internal tides. It is therefore obvious from the beginning that the simulation LSTIDE will not capture the increased vertical mixing caused by breaking internal tides. Hence, it is pointless to compare it to CVTIDE, which is constructed to capture precisely this effect. I therefore do not recommend publication.

Nevertheless, there are some interesting results, in particular the strengthened upwelling in the north Pacific in LSTIDE, and its effect on the global overturning and the hydrography. In the new version it is shown that this is not caused by increased vertical diffusivity $K_v$, but by increased resolved vertical buoyancy flux $\overline{w'b'}$ in the north Pacific. However, it is still not clear how the tidal motion leads to this buoyancy flux. It is speculated that it is caused by trapped tidal waves at the Kuril-Aleutian Ridge, but no specific support for this is presented. For example, an alternative hypothesis is that propagating internal tides with semidiurnal frequency are involved, since they are in fact resolved in at least part of this region, according to Fig. 15. It should be easy to check this by studying the tidal motion in detail, since the trapped internal tides have diurnal frequency. But even if it shown what kind of tidal waves are involved, it remains to understand why whey increase the vertical buoyancy flux.

A manuscript concentrating on these aspects might be publishable. The simulation CVTIDE should then be omitted. It cannot be expected to capture the effect of trapped waves, since its parameterization of the vertical diffusivity is constructed from the scaling of the generation of propagating

internal tides.

Such a manuscript would be so vastly different from the present one (with a very different title) that it should be considered as a new submission, not a revised one.

Detailed comments:

1. The language is sometimes incorrect, and needs to be checked. Some corrections are listed below.

2. There are many lengthy descriptions of figures that add little to what the reader can see for himself. Such descriptions are boring to read, and should be shortened strongly. There are also too many figures. Concentrate on the features that have a clear connection to the main conclusions!

3. Page 5, line 128: 'rate of dissipated mechanical energy' should be 'fraction of dissipated mechanical energy'.

4. Page 5, line 144: 'consists with the Prandtl number' should be 'agrees with the Prandtl number'.

5. Page 7, line 207: 'temperature biases' should be 'temperature differences'.

6. Page 9, line 265: 'the depth where the density over depth differs by 0.125 sigma units'. I don't understand this.

7. Page 9, line 279: 'enhanced for 4 Sv' shuld be 'enhanced by 4 Sv'.

8. Page 9, line 281: 'upwelling for 5 Sv' shuld be 'upwelling by 5 Sv'.

9. Page 10, line 308: 'surface pressure power' is better described as 'conversion from kinetic energy to barotropic potential energy'.

10. Page 10, line 310: 'the horizontal distributions show differences', and Page 12, line 356, 'higher buoyancy flux than NOTIDE': According to Fig. 14 h,i the differences are a factor $10^{-6}$ smaller than the values for NOTIDE, and hence insignificant.

11. Page 12, line 359 and Fig. 16: Since the buoyancy flux has different sign in different locations, the 'equivalent diffusivity' is negative in many places. Therefore, the interpretation of the buoyancy flux in terms of equivalent diffusivity does not make sense. However, this

interpretation is not needed, and can simply be omitted. What matters for the overturning is the diapycnal buoyancy flux, and the importance of the diffusivity is just that it causes such a flux.

12. Page 12, line 367: 'Trapped internal tides ... can be simulated with LSTIDE'. Check this! Trapped waves have a well defined spatial scale that can be compared to the model resolution, thus extending the analysis in Fig. 15 to the gray areas.

13. Page 13, lines 391-392: 'the upwelling in the North Pacific and Indonesian Archipelago links to the North Atlantic'. I don't think that this has been shown clearly, even though I agree that the alternative explanation by the Atlantic mixing can be ruled out. If there is such a link, why does it not hold in CVTIDE?

14. Page 13, line 395: 'the barotropic streamfunction has an offset effect in the vertical direction'. I don't understand what is meant by this.

15. Page 16, lines 481-482: The text seems to indicate that the increased ITF is caused by the increased upwelling in the north Pacific. A similar impression is given on page 12, line 362. However, the connection has not been shown. Perhaps the increased ITF is mainly caused by increased buoyancy flux and upwelling in the Indonesian Archipelago?

16. Figure 12: The units of the color scales should be given. It is also very unclear to me what is meant by 'difference of scaled vertical diffusivity'. It is strange that the differences seem to be several orders of magnitude larger than the values in NOTIDE.

17. Figure 14: The units of the color scales should be given. The sign of the buoyancy flux seems mostly positive in the plots, which contradicts the negative value in Table 3. Does the viscous dissipation include both horizontal and vertical viscosity?

18. Figure 15: Give the explicit expression for the plotted ratio.

---

## Author Response (AR2)

**Authors' Response to Reviews of**

**The tidal effects in the Finite-volumE Sea ice–Ocean Model (FESOM2.1): a comparison between parameterised tidal mixing and explicit tidal forcing**

Pengyang Song et al.
*Geosci. Model Dev. Discuss.,* `10.5194/gmd-2022-25`
* * *
**RC:** *Reviewers' Comment*,     AR: Authors' Response,     ☐ Manuscript Text

Dear editor and reviewers,

Thanks for the valuable comments on our manuscript. We treat all comments carefully, and list point-to-point responses below. Please also check the revised manuscript with tracked changes.

Yours sincerely,

Pengyang Song et al.

**1.   Reviewer #1 comments**

**RC:** *There are some significant improvements in the revised manuscript, but the main problem remains. Over most of the global ocean, the resolution of the model is insufficient to resolve the internal tides. It is therefore obvious from the beginning that the simulation LSTIDE will not capture the increased vertical mixing caused by breaking internal tides. Hence, it is pointless to compare it to CVTIDE, which is constructed to capture precisely this effect. I therefore do not recommend publication.*

*Nevertheless, there are some interesting results, in particular the strengthened upwelling in the north Pacific in LSTIDE, and its effect on the global overturning and the hydrography. In the new version it is shown that this is not caused by increased vertical diffusivity $K_v$ , but by increased resolved vertical buoyancy flux $\overline{w'b'}$ in the north Pacific. However, it is still not clear how the tidal motion leads to this buoyancy flux. It is speculated that it is caused by trapped tidal waves at the Kuril–Aleutian Ridge, but no specific support for this is presented. For example, an alternative hypothesis is that propagating internal tides with semidiurnal frequency are involved, since they are in fact resolved in at least part of this region, according to Fig. 15. It should be easy to check this by studying the tidal motion in detail, since the trapped internal tides have diurnal frequency. But even if it shown what kind of tidal waves are involved, it remains to understand why they increase the vertical buoyancy flux.*

*A manuscript concentrating on these aspects might be publishable. The simulation CVTIDE should then be omitted. It cannot be expected to capture the effect of trapped waves, since its parameterization of the vertical diffusivity is constructed from the scaling of the generation of propagating internal tides.*

*Such a manuscript would be so vastly different from the present one (with a very different title) that it should be considered as a new submission, not a revised one.*

**AR:** Thank you for your comments. The mixing in the North Pacific is not caused by the semidiurnal internal tides. Because the strength of a baroclinic tide is related to the respective barotropic tide, and the diurnal barotropic tides are much stronger than the semidiurnal ones in the Kuril–Aleutian Ridge (Fig. 1). We can also see from the model output that all signals in the Kuril–Aleutian Ridge show more significant diurnal

periods than semidiurnal periods (take one point as an example, Fig. 2).

In our control run, the mesh resolution is eddy-parameterising, and the model has no tides. Thus the model results roughly follow the geostrophic balance. Once we add global tides, we also add ageostrophic flows. Ageostrophic flows, such as tides, can easily cross the isobaths, therefore generating upwelling/downwelling around topographies. That is why LSTIDE increase mixing via turbulent vertical buoyancy flux but not $k_v$.

We do not agree to delete the CVTIDE part. Because we think comparing LSTIDE with CVTIDE would fill the gap between real tidal mixing and tidal mixing parameterisation. In climate research, a simulation of real tides is not applicable. Thus, improving the parameterisation is the only way to get a better diapycnal mixing in the model. From a climate study's point of view, the comparison is necessary.

**RC:** *The language is sometimes incorrect, and needs to be checked. Some corrections are listed below.*

AR: Thank you for the correction. We carefully treated all the comments and some similar grammatical mistakes in our manuscript.

**RC:** *There are many lengthy descriptions of figures that add little to what the reader can see for himself. Such descriptions are boring to read, and should be shortened strongly. There are also too many figures. Concentrate on the features that have a clear connection to the main conclusions!*

AR: Thank you for this suggestion. We delete some redundant sentences, figures and tables in the revised manuscript.

**RC:** *Page 5, line 128: 'rate of dissipated mechanical energy' should be 'fraction of dissipated mechanical energy'.*

AR: Thank you. This is corrected in the revised manuscript.

**RC:** *Page 5, line 144: 'consists with the Prandtl number' should be 'agrees with the Prandtl number'.*

AR: Thank you. This is corrected in the revised manuscript.

**RC:** *Page 7, line 207: 'temperature biases' should be 'temperature differences'.*

AR: Thank you. This is corrected in the revised manuscript.

**RC:** *Page 9, line 265: 'the depth where the density over depth differs by 0.125 sigma units'. I don't understand this.*

AR: Thank you for pointing out this. To make this clear, we rewrite this sentence as below.

> The second kind of MLD, following Monterey and Levitus (1997), is defined as the depth  where potential density is higher than the local sea surface density by 0.125 $kg\ m^{-3}$ .

**RC:** *Page 9, line 279: 'enhanced for 4 Sv' should be 'enhanced by 4 Sv'.*

AR: Thank you. This is corrected in the revised manuscript.

**RC:** *Page 9, line 281: 'upwelling for 5 Sv' should be 'upwelling by 5 Sv'.*

AR: Thank you. This is corrected in the revised manuscript.

**RC:** *Page 10, line 308: 'surface pressure power' is better described as 'conversion from kinetic energy to*

*barotropic potential energy'.*

AR:   Thank you. This is corrected in the revised manuscript.

RC:   *Page 10, line 310: 'the horizontal distributions show differences', and Page 12, line 356, 'higher buoyancy flux than NOTIDE': According to Fig. 14 h,i the differences are a factor $10^{-6}$ smaller than the values for NOTIDE, and hence insignificant.*

AR:   Thank you for pointing out this. In the Fig. 14 of the manuscript, panels (g), (h) and (i) are plotted in linear scale, not in decimal logarithmic scale. So the difference is not insignificant. To avoid misunderstanding, we modify this figure (see Fig. 3). The differences in buoyancy flux (Fig. 3b and 3c) are in the same order of magnitude compared with the original values in NOTIDE (Fig. 3a).

      However, the new plot does not show consistent positive patterns in the Indonesian Archipelago and Kuril–Aleutian Ridge. Table 1 shows that the regional integrated buoyancy flux in LSTIDE and CVTIDE are stronger than NOTIDE in these two areas. The regional integration results are also added to the revised manuscript.

RC:   *Page 12, line 359 and Fig. 16: Since the buoyancy flux has different sign in different locations, the 'equivalent diffusivity' is negative in many places. Therefore, the interpretation of the buoyancy flux in terms of equivalent diffusivity does not make sense. However, this interpretation is not needed, and can simply be omitted. What matters for the overturning is the diapycnal buoyancy flux, and the importance of the diffusivity is just that it causes such a flux.*

AR:   Thank you for this comment. We agree with your suggestion to delete the interpretation of 'equivalent diffusivity'. The turbulent buoyancy flux is shown in Fig. 3. This is corrected in the revised manuscript.

RC:   *Page 12, line 367: 'Trapped internal tides ... can be simulated with LSTIDE'. Check this! Trapped waves have a well defined spatial scale that can be compared to the model resolution, thus extending the analysis in Fig. 15 to the gray areas.*

AR:   Thank you for pointing out this. Following Falahat and Nycander (2015), we extend the plot to areas beyond the critical latitudes (Fig. 5). Even though trapped internal tides do not propagate as waves, they are confined in a horizontal scale (Falahat and Nycander, 2015). Figure 5 shows that the mesh applied in this work is fine enough to simulate trapped internal tides in the Kuril–Aleutian Ridge. This explanation is added to the revised manuscript.

      Falahat, Saeed, and Jonas Nycander. "On the generation of bottom-trapped internal tides." Journal of Physical Oceanography 45.2 (2015): 526-545.

RC:   *Page 13, lines 391–392: 'the upwelling in the North Pacific and Indonesian Archipelago links to the North Atlantic'. I don't think that this has been shown clearly, even though I agree that the alternative explanation by the Atlantic mixing can be ruled out. If there is such a link, why does it not hold in CVTIDE?*

AR:   Thank you for this comment. The linkage between the Indo-Pacific circulation and the Atlantic circulation is the Agulhas Current (Fig. 6). We can find from the barotropic streamfunction difference that there is a connection of the blue pattern between the Indian Ocean and the South Atlantic Ocean in Fig. 6c, but the connection is missing in Fig. 6b. The vector plots of barotropic velocity (around 25° E, 35° S) also show a net transport to the west occurring in Fig. 6c, but to the east in Fig. 6b. The plots indicate that the Agulhas Current is strengthened in LSTIDE but not in CVTIDE, which further alters the strength of the South Atlantic currents and the AMOC. Our results agree with the previous research that the Agulhas leakage has impacts the

MOC (Biastoch et al., 2008). For a better understanding of the cross-basin linkage, we add this explanation to our revised manuscript.

Biastoch, Arne, Claus W. Böning, and J. R. E. Lutjeharms. "Agulhas leakage dynamics affects decadal variability in Atlantic overturning circulation." Nature 456.7221 (2008): 489-492.

**RC:** *Page 13, line 395: 'the barotropic streamfunction has an offset effect in the vertical direction'. I don't understand what is meant by this.*

AR: Thank you for pointing out this. To make this clear, we rewrite this sentence as below.

> But note here  the vertical structure of oceanic circulation cannot be revealed by the barotropic streamfunction .

**RC:** *Page 16, lines 481–482: The text seems to indicate that the increased ITF is caused by the increased upwelling in the north Pacific. A similar impression is given on page 12, line 362. However, the connection has not been shown. Perhaps the increased ITF is mainly caused by increased buoyancy flux and upwelling in the Indonesian Archipelago?*

AR: Thank you for this comment. The streamfunction in the Fig. 11f of the manuscript shows the two upwelling regions in LSTIDE: the Kuril–Aleutian Ridge and the Indonesian Archipelago. Focusing on the upper 2000 m of the plot, we can find that the increase of ITF comes from both local (Indonesian Archipelago) upwelling and remote (Kuril–Aleutian Ridge) upwelling. By roughly estimating from the Fig. 11f of the manuscript, half of the increased ITF originates from the North Pacific cell, and the other half belongs to the South Indo-Pacific cell. Thus, the increased ITF is not only caused by the upwelling in the North Pacific. To avoid this misunderstanding, we revise the sentence as below.

> Our results show that the tidal mixing in these two areas drives stronger upwelling in the deep Pacific Ocean. In the Kuril–Aleutian Ridge, deep water is upwelled to the intermediate layer and advected to the tropical region; in the Indonesian Archipelago,  deep water is also upwelled to the intermediate layer and advected to the further south. Both upwelled water contributes to the ITF enhancement .
>
> The strong mixing upwells an additional 5 Sv of deep water in the North Pacific (around 50° N) to the intermediate layer and then spreads to the south (Fig. 11f).  The strong mixing in the Indonesian Archipelago also results in strong upwelling. The upwelled water from both the Indonesian Archipelago and Kuril–Aleutian Ridge contributes to the ITF (Fig. 13c) and flows into the Indian Ocean .

**RC:** *Figure 12: The units of the color scales should be given. It is also very unclear to me what is meant by 'difference of scaled vertical diffusivity'. It is strange that the differences seem to be several orders of magnitude larger than the values in NOTIDE.*

AR: Thank you for pointing out this. What we wanted to show is that the deep-ocean vertical diffusivity in CVTIDE is one to two orders larger than NOTIDE.

This is true in the model. In NOTIDE, the KPP scheme provides a background vertical diffusivity of $10^{-5}m^2s^{-1}$, and the shear-induced diffusivity is small due to the low velocity in the deep ocean. In CVTIDE, the deep-ocean vertical diffusivity contains an additional component from the CVMIX_TIDAL parameterisation, which provides a vertical diffusivity up to $0.5 \times 10^{-2}m^2s^{-1}$. Comparing the two values,

one can find that the vertical diffusivity in CVTIDE is at most three orders of magnitude larger than the values in NOTIDE. But if we calculate the global average, this difference is about one order of magnitude, as shown by Simmons et al. (2004). Since tidal mixing is bottom-enhanced, we change the approach to show the vertical diffusivity, and the plot is modified as Fig. 7.

Simmons, Harper L., et al. "Tidally driven mixing in a numerical model of the ocean general circulation." Ocean Modelling 6.3-4 (2004): 245-263.

**RC:** *Figure 14: The units of the color scales should be given. The sign of the buoyancy flux seems mostly positive in the plots, which contradicts the negative value in Table 3. Does the viscous dissipation include both horizontal and vertical viscosity?*

AR:   Thank you for pointing out this. To avoid misunderstanding, we leave the decimal logarithmic scale only in the colorbars. Also, the units are given in the plots.

Besides, we found two mistakes in our plotting routine. One mistake is that we used natural logarithms (log) instead of decimal logarithms (log10), that causes the errors in plotting the bottom drag and viscous dissipation. The second mistake is that we used filled contour plots (tricontourf) instead of pseudocolor plots (tripcolor). Filled contour plots seem to overlook the scattered patterns, which is why there seem to be no negative values in the Figs. 14h and 14i of the manuscript.

In the Figs. 14d, 14e and 14f of the manuscript, we plotted only vertical viscous dissipation. Since horizontal viscous dissipation is diagnosed in our manuscript, the revised figure includes both horizontal and vertical viscous dissipation. We mention this in our revised manuscript.

To sum up, the Fig. 14 of the manuscript is revised as Fig. 3 and Fig. 4 below.

**RC:** *Figure 15: Give the explicit expression for the plotted ratio.*

AR:   Thank you for this comment. Explicit expressions are added to the revised manuscript.

**2.   Reviewer #2 comments**

**RC:** *Upon request from the topical editor, the revised version of the manuscript has been reviewed in view of the interactive public discussion. I should report on this assessment. In the light of a criteria whether the authors responded appropriately to the reviewers' original concerns or not, I think that the authors have made a basically good response. But I think some additional work is required before the paper can be accepted for final publication.*

AR:   Thank you for your comments. Point-to-point responses are listed below.

**RC:** *The second comment of Reviewer #1 and the first comment of Reviewer #2 commonly concern the reality of the sink for the long internal tidal waves in LSTIDE experiment with explicit tides. Authors seems to have reached a conclusion that the convergence of vertical turbulent buoyancy flux due to tides balances the vertical advection term for the mean field, contrary to their first thought that the vertical mixing in LSTIDE experiment would be enhanced by vertical shear of baroclinic tides. This may not be realistic as the reviewers have pointed out, but would be reasonable given the spatial resolution of the model being used here. I think the novelty and value of this paper is to have presented how the meridional overturning circulations would change in a simulation with explicit tides. For the paper to set a useful basis for future works with high-resolution models and to have a lasting value, two-dimensional map of energy sink for*

*LSTIDE experiment should be desirably compared with one of state-of-the-science estimates such as the one presented by de Lavergne et al. (2019) and differences are discussed.*

AR: Thank you for this comment. Comparing results from a tide model with de Lavergne et al. (2019) would be interesting, but we think it is unnecessary for this work. The low-resolution mesh we applied is for climate simulations, thus (1) the model does not generate enough baroclinic tidal energy; (2) the model cannot describe high-mode internal tides; (3) the model cannot simulate wave-wave interactions. Such comparison can only be made by using a much higher resolution, such as $0.1°$ (Li et al., 2015), but then we cannot simulate a long-term ocean state due to the limitation of computational capacity. We agree that some work on comparing simulation and theoretical results would be valuable, but not much related to our focus.

de Lavergne, Casimir, et al. "Toward global maps of internal tide energy sinks." Ocean Modelling 137 (2019): 52-75.

Li, Zhuhua, Jin-Song Von Storch, and Malte Müller. "The M2 internal tide simulated by a 1/10° OGCM." Journal of Physical Oceanography 45.12 (2015): 3119-3135.

RC: *The globally integrated kinetic energy budget for the three experiments is presented in Table 3. The value for the buoyancy flux term is the same for all experiments. I think more explanation for this feature would be useful.*

AR: Thank you for mentioning this. The values of global integration are close, but not identical (because only the first two decimal places are kept). We add regional integration results in Table 1. By integrating over the Kuril–Aleutian Ridge (120° E–120° W, 30° N–60° N) and Indonesian Archipelago (90° E–150° W, 15° S–15° N), we can see that the values of regional integration are different. Note here negative values indicate upwelling. For comparison, we also give the values of regional integration over the North Atlantic (60° W–10° E, 45° N–70° N), which are positive values. It is also comprehensible that LSTIDE shows larger positive values in the North Atlantic due to the stronger AMOC.

RC: *What is the source of the temporal mean square tidal velocity distribution used for CVTIDE experiment? How different are the distribution used for CVTIDE and that of LSTIDE? Or are they common?*

AR: Thank you for this comment. The forcing file used in CVTIDE is interpolated from the file used in Jayne (2009), originating from Jayne and St. Laurent (2001). The forcing file only consists of energy dissipation, which is calculated via Eq. (3) in the manuscript. We cannot find the original temporal mean square tidal velocity distribution from the file.

However, to our knowledge, the tidal model Jayne (2009) used is driven by the largest eight tidal components. In LSTIDE, we use the lunisolar gravitational potential, which consists of all tidal components theoretically. But note that except for the largest eight tidal components, the rest tidal constituents are minor.

Since we do not have the output from the tidal model of Jayne (2009), we cannot directly compare. But we assume both tidal models are valid, so the temporal mean square tidal velocity distribution should be close. The validation of the tidal potential module in our work is shown in Section 3.

Jayne, Steven R., and Louis C. St. Laurent. "Parameterizing tidal dissipation over rough topography." Geophysical Research Letters 28.5 (2001): 811-814.

Jayne, Steven R. "The impact of abyssal mixing parameterizations in an ocean general circulation model." Journal of Physical Oceanography 39.7 (2009): 1756-1775.

Table 1: Regional integrated buoyancy flux in three typical areas. Buoyancy flux is expressed as $\rho_0 bw$.

| Experiment ID | Kuril–Aleutian Ridge | Indonesian Archipelago | North Atlantic |
|---|---|---|---|
| NOTIDE | $-1.22$ | $-0.82$ | 0.38 |
| CVTIDE | $-1.52$ | $-1.13$ | 0.39 |
| LSTIDE | $-2.78$ | $-1.19$ | 0.49 |

[Figure]

Figure 1: Co-tidal charts of the four main tidal components from the tide model. The colours indicate tide amplitudes, and the white lines indicate Greenwich phase lags with an interval of $60°$.

[Figure]

Figure 2: Vertical velocity ($w$) and potential density ($\rho$) from one-point hourly-output results. The model point locates at 1920 m in 165° E, 55° N. Potential density is model potential density minus a constant value (1030 $kg/m^3$).

[Figure]

Figure 3: Horizontal distribution of vertically integrated buoyancy flux terms. Buoyancy flux is expressed as $g\rho w$, while turbulent buoyancy flux is $\overline{g\rho' w'}$ (positive values represent upwelling). The left column shows the control run, and the middle and right columns show the differences.

[Figure]

Figure 4: Horizontal distribution of vertically integrated energy dissipation terms in Table 3 of the manuscript (positive values represent energy dissipation). The upper and lower rows indicate bottom drag and viscous dissipation (including horizontal and vertical components). The left column shows the control run, and the middle and right columns show the differences. Note that the plots are shown in a decimal logarithmic scale.

[Figure]

Figure 5: The ratio of mode-1 K1/M2 internal tide scale and mesh resolution. The thick grey dashed line indicates the critical latitudes of internal tides.

[Figure]

Figure 6: The barotropic streamfunction and velocity in the three sensitivity runs. The shading plot shows the barotropic streamfunction, which is the same as Figure 13 in the manuscript but zoom in the Agulhas current region. The vector plot shows the barotropic velocity. The upper panel shows the control run, while the middle and lower panels show the differences.

[Figure]

Figure 7: The left panels are horizontal maps of depth-averaged vertical diffusivity over the bottom 500 m (average over the whole depth where bathymetry is less than 500 m). The right panel shows the global mean vertical diffusivity profile in the three sensitivity runs. Note that colorbars are shown in decimal logarithmic scales.